# Protection of Aluminum Alloy 3003 in Sodium Chloride and Simulated Acid Rain Solutions by Commercial Conversion Coatings Containing Zr and Cr

**Maja Mujdrica Kim** [1,2], **Barbara Kapun** [3], **Urša Tiringer** [3], **Gavrilo Šekularac** [2,3] and **Ingrid Milošev** [3,*]

1    TRIMO d.o.o., Prijateljeva c. 12, SI-8210 Trebnje, Slovenia
2    Jožef Stefan International Postgraduate School, Jamova c. 39, SI-1000 Ljubljana, Slovenia
3    Department of Physical and Organic Chemistry, Jožef Stefan Institute, Jamova c. 39, SI-1000 Ljubljana, Slovenia
*    Correspondence: ingrid.milosev@ijs.si; Tel.: +386-1-4773-452

**Abstract:** The morphology, composition and corrosion properties of commercial hexafluoro-zirconate trivalent chromium coatings (SurTec® 650) deposited on chemically cleaned aluminum alloy 3003 were studied. The coatings were deposited at room temperature using different concentrations of SurTec® 650 (10, 25 and 50 vol.%) and different conversion times (90 s, 11 min and 18 min). Scanning electron microscopy with energy dispersive X-ray spectrometry, X-ray photoelectron spectroscopy and time-of-flight secondary ion spectrometry were employed to investigate the surface morphology, composition and thickness of uncoated and coated AA3003 samples. The morphology of the coating varied from uniform nodular to non-uniform and cracked; coatings were deposited at intermetallic particles and at the alloy matrix. The main constituents of conversion coatings were Zr(IV) and Cr(III) oxides; in addition to oxides, fluorides were also formed. The corrosion properties were investigated in two solutions: more aggressive sodium NaCl and less aggressive simulated acid rain. These commercial conversion coatings exhibited a good corrosion resistance but only after longer immersion in solution, i.e., 24 h. The results reveal an interesting behavior of zirconate-based coatings on aluminum-manganese alloy.

**Keywords:** aluminum alloy 3003; trivalent chromium process (TCP); hexafluoro zirconate conversion coating; simulated acid rain; sodium chloride; X-ray photoelectron spectroscopy

## 1. Introduction

Chemical conversion coatings are widely used on aluminum-based alloys and other technologically important materials with the aim to protect the metal against corrosion and to promote adhesion of the primer coating to the substrate. Conversion coatings are formed by immersing a metallic substrate in a chemical bath. The reaction between the metal surface and the components in the chemical solution results in the growth of a coating layer on the surface. The coating becomes an integral part of the surface, as there is no mechanical interface between the coating and the base material. One of the most important conversion coatings in the past were chromate conversion coatings (CCCs). Due to high toxicity and health concerns related to hexavalent chromium, numerous studies are currently devoted to alternatives for the replacement of Cr(VI)-based surface treatments. These include chemical conversion coatings, organic sol-gel coatings, nanocomposites, Cr(VI)-free anodization, metal rich primers, etc. [1–3]. Preferably, non-chromate treatment should provide protection that is equivalent or

better than the hexavalent conversion coating. Currently, chemical conversion coatings are the only commercial alternative to CCCs [4]. Trivalent chromium conversion coatings, which we refer to in this study, are one of the chemical conversion coatings available.

The Naval Air Systems Command (NAVAIR) has formulated a new trivalent chromium process (TCP) coating, totally free of Cr(VI) [5–8]. It generally contains hexafluoro zirconic acid and a small amount (≤5 wt.%) of trivalent chromium salt and forms a Zr-/Cr-rich oxide coating when deposited on a substrate [5–8]. Some commercial substitute systems have been well implemented and TCP, as one of the leading non-chromate conversion coatings, has provided excellent corrosion protection for aluminum alloys [5–8]. The formation of a small amount of Cr(VI) was reported to occur during TCP in oxygenated solution due to the formation of $H_2O_2$ and its ability to oxidize Cr(III) to Cr(VI) [9].

The mechanism of formation of TCP coatings on aluminum alloys (AA) AA2024 [10], AA6061 [8] and AA7075 [8] appears to be similar for all three alloys. After the initial attack on the surface oxide film with $H^+$ and/or $F^-$ ions, the cathodic reaction of oxygen reduction at localized intermetallic sites causes an increase in the interfacial pH that drives the formation of zirconium-chromium mixed ($ZrO_2 \cdot 2H_2O$ and $Cr(OH)_3$) oxide in the outer layer and aluminum oxide at the metal/coating interface. Guo and Frankel reported that a two-layered coating structure was formed on AA2024 [7].

Chen et al. [11] investigated the formation of TCP coatings on AA5052 as a function of immersion time in a conversion bath. An immersion period of 30–300 s represents the main stage of coating growth, as determined by the open-circuit potential (OCP) vs. time curves. Coatings formed for 300 s exhibited a uniform structure, whilst after 600 s, the coating surface was cracked [11]. Cracking of the coating was ascribed to the dehydration of the TCP coating during exposure to high vacuum [7]. The coatings consisted mainly of Cr, Al and Zr oxides and fluorides [7].

Studies of TCP coatings on different aluminum alloys showed that the protection mechanism differs depending on the type of aluminum and type of corrosion medium (chloride and non-chloride containing electrolytes). Li et al. [8] studied electrochemical properties on TCP-coated AA6061 and AA7075 during full immersion tests at room temperature in air-saturated $Na_2SO_4$ and in a more aggressive mixture of $Na_2SO_4$ and NaCl. The coating on AA6061 acted primarily as a cathodic inhibitor, while on AA7075, it provided anodic and cathodic protection. The coating was efficient even in the presence of chloride, as no pitting was detected on AA6061 at potentials up to 0.5 V vs. Ag/AgCl, and with pitting potential being shifted more positively by ~400 mV, i.e., up to 0.074 V vs. Ag/AgCl for AA7075.

On AA2024 the TCP coating provided both anodic and cathodic protection, similarly to AA7075, which was ascribed to physical blockage of Al-rich sites (where oxidation takes place) and Cu-rich intermetallic sites (where reduction takes place) [10]. TCP coatings on AA2024-T3 have also been studied by Guo and Frankel [7]. Corrosion measurements were made in dilute Harrison's solution (0.05 wt.% NaCl + 0.35 wt.% $(NH_4)_2SO_4$) which is much less aggressive than pure chloride solutions such as 0.5 M NaCl. The TCP coating provided corrosion protection to AA2024-T3 by increasing the breakdown potential and suppressing the oxygen reduction reaction on the surface of the aluminum alloy [7].

In comparison to other aluminum alloys, relatively little is known about the electrochemical performance of conversion coatings on AA3003. Banczek et al. [12] compared the effect of a cerium conversion coating, self-assembling molecules (SAM) treatment, and a combination of these two treatments on the corrosion resistance of the AA3003 alloy. The results were compared with that obtained for the SurTec 650 Chromital TCP conversion coating containing hexavalent chromium. The results demonstrated that the chromate layer promoted the highest corrosion resistance among the tested surface treatments, and the next highest corrosion resistance was found for the SAM treatment. Smit et al. [13] studied the performance and characteristics of a titanium-based conversion coating on aluminum alloy 3003. The $H_2TiF_6$ based conversion coating on AA3003 improved anodic inhibition and reduced the corrosion current density. No cathodic inhibition was noticed.

To ensure good corrosion protection, not only is the application of appropriate conversion coating important but the cleaning and surface preparation of the substrate also have to be considered. Alkaline cleaning is widely used for removal of oils, organic contaminants, and naturally grown oxide layers. The smut layer formed during alkaline cleaning and insoluble intermetallic particles are then removed using $HNO_3$ which also passivates the surface [14]. Guo and Frankel noted that although the pre-treatment could significantly affect the surface morphology and chemical composition, the resulting TCP coating was similar [7].

In this research, we need to address the knowledge gap on TCP coatings deposited on AA3003. This wrought aluminum-manganese alloy is commonly used for roofing, gutters, siding, etc. It exhibits moderate strength, good workability and good corrosion resistance. Herein, we report on the morphology, composition and electrochemical properties of TCP-coated AA3003. A commercial TCP coating—SurTec650—hexafluoro-zirconate bath with the addition of trivalent chromium—was investigated. Coatings were prepared at different bath compositions and conversion times. Electrochemical measurements, scanning electron microscopy (SEM) with energy dispersive X-ray spectrometry (EDS), X-ray photoelectron spectroscopy (XPS) and time-of-flight secondary ion spectrometry (ToF-SIMS) were employed to investigate the corrosion properties and surface morphology, composition and thickness of AA3003 after immersion in a TCP conversion bath. The corrosion properties were investigated in two solutions: more aggressive NaCl and less aggressive simulated acid rain. Both solutions are relevant for applications of AA3003 in various environments.

## 2. Materials and Methods

### 2.1. Materials and Chemicals

For this study, the 50 micrometer-thick foil of aluminum alloy EN AW 3003 H19 (denoted as AA3003) produced by Impol-TLM d.o.o. was used (Šibenik, Croatia). The chemical composition of the alloy is Mn 1.1 wt.%, Fe 0.6 wt.%, Si 0.14 wt.%, Cu 0.13 wt.%, Zn 0.009 wt.%, Cd 0.0003 wt.%, and Al 98.1 wt.%, as specified by the manufacturer. The foil was cut into specimens of dimensions 20 mm × 40 mm.

The following chemicals were used for surface pre-treatment: absolute ethanol (Carlo Erba reactants, Rodano, Italy), SurTec® 132 and SurTec® 089 (SurTec International GmbH, Bensheim, Germany) cleaning agents and nitric acid (65%, Sigma Aldrich, Saint Louis, MO, USA). The major component of the slightly alkaline builder SurTec® 132 is tetrapotassium pyrophosphate. It is free of silicates and surfactants, with a pH of 8.3. Recyclable soak detergent SurTec® 089 contains non-ionic surfactant alcohols such as amines, coco alkyl and ethoxylated fatty alcohol. Milli-Q Direct water with a resistivity of 18 MΩ cm at 25 °C (Millipore, Billerica, MA, USA) was used for rinsing and solution preparation.

Commercially available SurTec® 650 (SurTec International GmbH), a hexafluoro-zirconate type of conversion coating with the addition of trivalent chromium Cr(III) was used for the preparation of coatings.

Electrochemical measurements were performed in two different solutions: 3.5 wt.% NaCl (99.7%, Fisher Scientific, Hampton, NH, USA), pH~5.5, and simulated acid rain (0.2 g $Na_2SO_4$ (99%, Acros Organics, Hampton, NH, USA) + 0.2 g $NaHCO_3$ (Sigma Aldrich, Saint Louis, MO, USA) + 0.2 g $NaNO_3$ (99%, Sigma Aldrich, Saint Louis, MO, USA) in 1 L). The pH of the solution was adjusted to 4.9–5.1 with the addition of 0.1 M $H_2SO_4$ (96%, Carlo Erba reagents, Barcelona, Spain).

### 2.2. Chemical Pre-Treatment

Before applying the coating, the metal surface was chemically pre-treated. Samples were initially cleaned in absolute ethanol (3 min) in an ultrasound bath (Elmasonic P 30 H, Singen, Germany) and dried in a stream of nitrogen. Chemical cleaning was then carried out using a mixture of SurTec® 132 and SurTec® 089, followed by desmutting of the surface with 50% $HNO_3$ in water, pH = 1.5.

The optimization of chemical pre-treatment was carried out to consider the following parameters: (i) the concentration of SurTec® 132 and of SurTec® 089, (ii) temperature (iii) immersion time in a mixture of SurTec® 132 and 089 (ST132/089), and (iv) immersion time in a $HNO_3$ solution. Details of the chemical pre-treatment are described in Appendix A.

### 2.3. Conversion Coating

Conversion coating SurTec 650 was deposited on the pre-treated metal samples. The samples were immersed in a SurTec® 650 solution bath (pH~3.9; if necessary adjusted with 1 wt.% NaOH or 5 wt.% $H_2SO_4$) in a 250-mL polyethylene cup. All coatings were prepared at room temperature. The solution was agitated using a magnetic stirrer at a speed of 250 rpm. Samples were then rinsed with 50 mL of Milli-Q water, immersed in it for 1 min at room temperature and finally, dried in a stream of nitrogen. The following parameters were considered: (i) the concentration of SurTec® 650, and (ii) conversion time (i.e., immersion time in the conversion bath).

### 2.4. Electrochemical Measurements

Electrochemical measurements were performed at room temperature in a three-electrode conventional corrosion cell (K0235 Flat Cell Kit, volume 250 mL). The specimen (working electrode) was embedded in a Teflon holder with an exposed area of 1 $cm^2$. An Ag/AgCl electrode was used as the reference electrode (potential 0.192 V vs. standard hydrogen electrode) and a platinum mesh served as the counter electrode.

Electrochemical experiments were carried out with the potentiostat/galvanostat Autolab PGSTAT 12 (Metrohm Autolab, Utrecht, Netherlands) and controlled by Nova 2.1 software. In this study, all the potentials are referred to the Ag/AgCl scale. First, the electrode was stabilized for 1 h at the open circuit potential to attain a quasi-equilibrium value denoted as OCP. Linear polarization measurements were then carried out in a potential range ±10 mV vs. OCP using a 0.1-mV/s potential scan rate. The polarization resistance ($R_p$) was determined from the slope of fitted current density vs. potential lines using Nova software. Afterwards, the potentiodynamic polarization measurements were performed using a 1-mV/s potential scan rate starting at −250 mV vs. OCP in the anodic direction. For each sample, measurements were repeated at least three times and the most representative measurement was chosen to be presented in graphs. The results in tabular form are shown as mean ± standard deviation. Electrochemical corrosion parameters, corrosion potential ($E_{corr}$), corrosion current density ($j_{corr}$), and breakdown potential ($E_{bd}$) were determined from polarization curves by Tafel approximation using Nova 2.1 software. The $E_{bd}$ was determined as a potential at which current density starts to increase abruptly within the passive range. The value of the passive region was denoted as $\Delta E = |E_{bd} - E_{corr}|$.

The mechanism of coating deposition was studied using OCP vs. time curves. The same equipment was used as mentioned above, with a 250-mL polyethylene cup as a cell. The exposed area of each specimen was 2 $cm^2$. An Ag/AgCl electrode was employed as the reference electrode. The measurements were repeated three times with similar results.

### 2.5. Surface Analysis

Scanning electron microscopy (SEM) images using the backscattered or secondary electron signal were recorded using a JEOL JSM-7600F microscope (Peabody, MA, USA) at 5 kV or 10 kV. The instrument is equipped with an energy dispersive X-ray spectrometer (EDS) supplied by Oxford Instruments INCA (Abingdon, UK). The selected analyzed area was a few micrometers in diameter and the analyzed depth was around 1 μm. The analysis was performed at an energy of 10 kV or 15 kV. Prior to analysis, the specimens were coated with a thin carbon layer.

X-ray photoelectron spectroscopy (XPS) was used to determine the chemical composition of the surface of a material using a PHI-TFA XPS spectrometer, produced by Physical Electronic Inc., Feldkirchen, Germany. During the analysis, the vacuum was in the range of $10^{-9}$ mbar. The analyzed area was 0.4 mm in diameter and surface sensitivity around 3–5 nm. The sample surface was excited

by X-ray radiation from a monochromatic Al source at a photon energy of 1486.6 eV. The survey wide-energy spectra were taken with a pass energy of the analyzer of 187 eV in order to identify and quantify elements present on the surface. The high-energy resolution spectra were acquired with an energy analyzer operating at a resolution of about 0.6 eV and a pass energy of 29 eV. XPS spectra were analyzed by Multipak software, version 8.1 (Physical Electronic Inc.). During data processing, the spectra were aligned by setting the C 1-s peak at 284.8 eV, characteristic for C–C/C–H bonds. Carbon as an adventitious element was not considered for the calculation of the surface elemental composition.

The fitting procedure allowed signals to be evaluated by determining peak position, height, width and Gaussian/Lorentzian ratio. The border conditions were fixed in accordance with parameters of standard peaks for metal and oxides. Details on the fitting procedure and standards used are given in Appendix B.

Time-of-flight secondary mass ion spectrometry (ToF-SIMS) ion depth profiles were made using a TOF.SIMIS 5 (ION TOF, Münster, Germany). Sputtering was carried out using a $Cs^+$ ion beam at 2 keV over rastering over 400 μm × 400 μm area. Spectra were recorded using a $Bi^+$ primary ion beam at 30 keV over a 100 μm × 100 μm area. Negative secondary ions were detected. A sputtering rate of ~0.35 nm/s was estimated relative to a Ni/Cr standard.

## 3. Results and Discussion

In the first part of the study, the effect of chemical pre-treatment on the morphology, composition and corrosion properties of AA3003 samples was analyzed, resulting in an optimized procedure. The latter was then used as a chemical pre-treatment prior to deposition of the SurTec® 650 conversion coating. The morphology, composition and corrosion properties of SurTec® 650-coated AA3003 samples were analyzed.

*3.1. Chemical Pre-Treatment of AA3003 Substrate*

3.1.1. Electrochemical Measurements

To optimize the cleaning pre-treatment using commercial cleaners SurTec® 132 and SurTec® 089 along with $HNO_3$ as a desmutting agent, the following parameters were considered: (i) the concentrations of SurTec® 132 and SurTec® 089 to select the concentration within the mixture SurTec 132/089, (ii) temperature and (iii)immersion time in SurTec® 132/089, and (iv) immersion time in the $HNO_3$ solution (Figure A1). The samples were prepared by keeping all parameters constant except one. For the samples prepared under different conditions, the potentiodynamic polarization curves in simulated acid rain were measured; electrochemical parameters—$R_p$, $E_{corr}$ and $j_{corr}$—served as decisive parameters for selecting the optimal condition. The potentiodynamic polarization curves measured in simulated acid rain are provided in the Appendix A, together with the deduced electrochemical parameters (Figures A2–A5, Table A1). Based on the difference between the treated and untreated samples, the following parameters were selected as the final chemical pre-treatment: a mixture of 2% ST 132 and 0.5% ST 089, an immersion time of 3 min, an immersion temperature of 40 °C and a desmutting time of 30 s at room temperature, denoted as ST/$HNO_3$.

The comparison of polarization curves recorded in 3.5% NaCl and simulated acid rain for the as-received untreated AA3003 sample and the sample after the selected chemical pre-treatment with ST/$HNO_3$ are presented in Figures 1 and 2. The related electrochemical parameters are shown in Table 1.

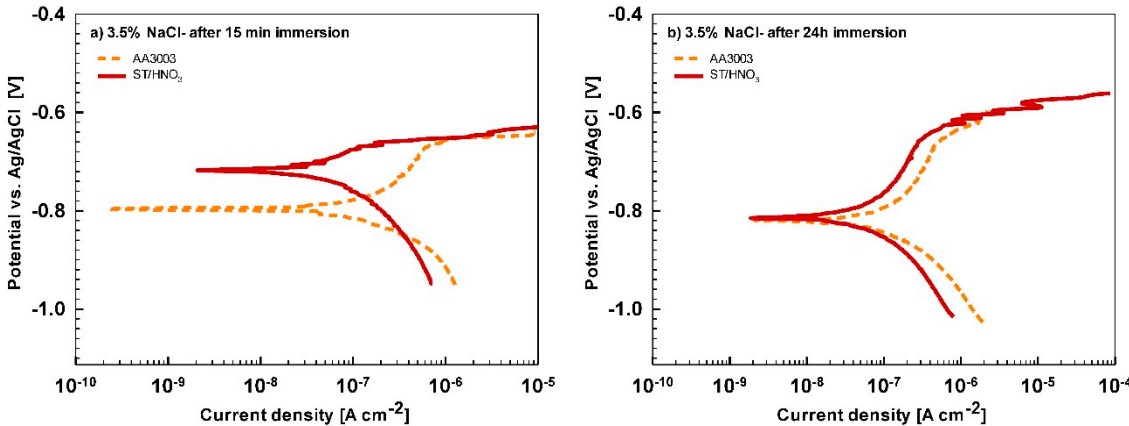

**Figure 1.** Potentiodynamic polarization curves recorded for the as-received AA3003 and the chemically pre-treated AA3003 (ST/HNO$_3$) in 3.5% NaCl after immersion times of (**a**) 15 min, (**b**) 24 h. The scan rate was 1 mV/s.

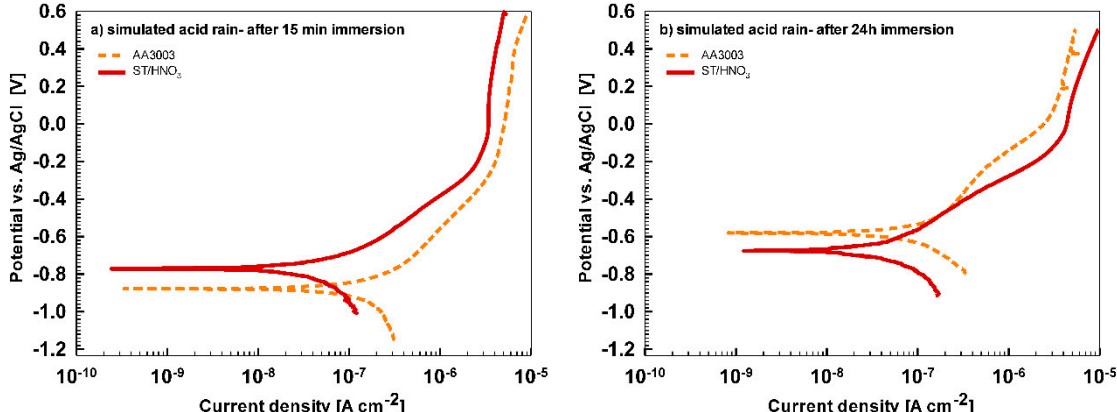

**Figure 2.** Potentiodynamic polarization curves recorded for as-received (bare) AA3003 and chemically pre-treated AA3003 (ST/HNO$_3$) in simulated acid rain after immersion times of (**a**) 15 min, (**b**) 24 h. The scan rate was 1 mV/s.

**Table 1.** Polarization resistance ($R_p$), corrosion current density ($j_{corr}$), corrosion potential ($E_{corr}$) and breakdown potential ($E_{bd}$) measured after different immersion times in 3.5% NaCl and simulated acid rain for the as-received AA3003 and the chemically pre-treated AA3003 (ST/HNO$_3$) (Figures 1 and 2). Values are given as mean ± standard deviation.

| Electrolyte | Immersion Time | Sample | $R_p$/k$\Omega$ cm$^2$ | $E_{corr}$/V | $j_{corr}$/µA cm$^{-2}$ | $E_{bd}$/V |
|---|---|---|---|---|---|---|
| 3.5% NaCl | 15 min | AA3003 | 561 ± 45 | −0.78 ± 0.01 | 0.10 ± 0.04 | −0.66 |
| | | ST/HNO$_3$ | 682 ± 85 | −0.74 ± 0.03 | 0.040 ± 0.003 | −0.67 |
| | 24 h | AA3003 | 315 ± 118 | −0.81 ± 0.02 | 0.10 ± 0.03 | −0.63 |
| | | ST/HNO$_3$ | 735 ± 299 | −0.80 ± 0.02 | 0.06 ± 0.03 | −0.66 |
| Simulated acid rain | 15 min | AA3003 | 610 ± 131 | −0.88 ± 0.003 | 0.09 ± 0.02 | - |
| | | ST/HNO$_3$ | 825 ± 229 | −0.76 ± 0.01 | 0.07 ± 0.02 | - |
| | 24 h | AA3003 | 596 ± 273 | −0.66 ± 0.11 | 0.30 ± 0.28 | - |
| | | ST/HNO$_3$ | 1527 ± 242 | −0.64 ± 0.05 | 0.04 ± 0.01 | - |

Figure 1a shows the polarization curves recorded for the as-received and chemically pre-treated AA3003 in 3.5% NaCl after 15 min immersion. Following $E_{corr}$, the increase in current density was not abrupt and $E_{bd}$ appeared at −0.66 V, giving a $\Delta E$ of 120 mV. This behavior indicates a relatively good corrosion resistance of AA3003 in chloride-containing solution compared to other, less

corrosion-resistant alloys [15]. Compared to the untreated AA3003, the chemically pre-treated samples showed smaller $j_{corr}$ and more positive $E_{corr}$ (Table 1). The improvement of the corrosion parameters is a consequence of the removal of the naturally grown layer and subsequent passivation in nitric acid. $E_{bd}$ was similar (−0.66 V and −0.67 V, respectively) for both samples.

After 24 h immersion in 3.5% NaCl (Figure 1b), the shape of the polarization curve for the untreated AA3003 was rather similar, but the passive range seemed to be more pronounced. In contrast, the shape of the curve for the chemically pre-treated sample changed considerably: $E_{corr}$ was more negative, resulting in an increase of $\Delta E$ from 70 mV to 140 mV. The $j_{corr}$ of both samples were similar, but the passive range established for the chemically pre-treated sample doubled in comparison to the short immersion time (Figure 1a).

Polarization curves recorded in simulated acid rain, which does not contain aggressive chloride ions, are presented in Figure 2. After being subjected to chemical pre-treatment, the AA3003 substrate showed a similar shape of the curve in simulated acid rain after 15 min immersion (Figure 2a) as the non-treated sample; however, $E_{corr}$ shifted positive, $j_{corr}$ was smaller and $R_p$ larger. After 24 h immersion (Figure 2b) in simulated acid rain, $j_{corr}$ for both samples was reduced and $E_{corr}$ became more positive.

The shapes of the curves recorded for AA3003 in NaCl (pH~5.5) and $Na_2SO_4$ + $NaHCO_3$ + $NaNO_3$ (simulated acid rain, pH 4.9–5.1) differed considerably: in simulated acid rain, no breakdown was observed in the anodic range where the current density reached a plateau region following the Tafel range (Figures 1 and 2). Different shapes reflect different electrochemical processes occurring in sulfate-containing solution (acid rain) and chloride-containing solution. In the presence of chloride, aluminum alloys are subjected to localized corrosion which starts at intermetallic particles, due to their different electrochemical activity, as compared to the aluminum alloy matrix [16,17]. The latter is also dependent on the pH of the solution [18]. In near-neutral solutions, aluminum is protected by the formation of aluminum hydroxide, $Al(OH)_3$, generally expressed as

$$Al + 3H_2O \rightarrow Al(OH)_3 + 3H^+ + 3e^- \tag{1}$$

Corrosion attack occurs when chloride ions are adsorbed on the weak points of the aluminum hydroxide film. One possible mechanism may involve the formation of intermediate soluble chloride complexes, according to Equation (2) [19]:

$$Al + nCl^- \rightarrow AlCl_n{}^{(n-3)-} + 3e^- \tag{2}$$

This process leads to thinning of the passive layer and pitting corrosion.

In near-neutral solution, which does not contain chloride ions, like dilute sulfate solution, a passive film is formed to protect the underlying substrate (Figure 2). Please note that the plateau following the Tafel range extended up to high anodic potentials, above 1.4 V, with no breakdown potential observed (*y*-axis in Figure 2 is given up to 0.6 V). Based on XPS analysis it was reported that the thickness of the protective layer increased by anodic polarization and its composition approached AlOOH ($Al_2O_3$ + $H_2O$) [20]. Presumably, sulfate ions are incorporated into the oxide layer; their incorporation would lead to the formation of a layer with reduced solubility and thus, increased protectiveness, as was shown recently by the synergistic effect of cerium and sulfate ions [21]. In addition to sulfate ions, simulated acid rain contains nitrate and carbonate ions; nitrates are known to be passivating agents [22] and in the presence of carbonate ions, an insoluble deposit may form, which retards the development of pitting corrosion [23].

### 3.1.2. SEM/EDS Characterization

The morphology and composition of the pre-treated sample was examined by SEM/EDS analyses at different locations on the sample surfaces. The morphology of the surface is presented in Figure 3,

and Table 2 presents a summary of the EDS analysis carried out at the denoted spots enumerated in Figure 3.

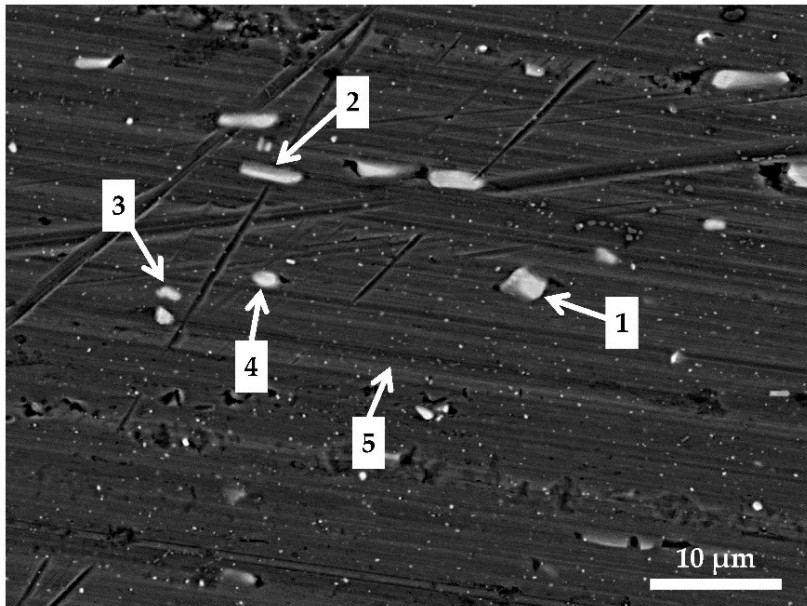

**Figure 3.** SEM backscattered electron micrograph of AA3003 chemically pre-treated using ST/HNO$_3$. The enumerated spots are the locations at which EDS analysis was performed (Table 2).

**Table 2.** The concentrations of elements obtained by EDS analysis at different locations on the chemically pre-treated sample with Surtec 132/089 and 50% HNO$_3$ (Figure 3).

| Location | at.% O | at.% Al | at.% Mn | at.% Fe | at.% Si | at.% Cu | at.% F |
|---|---|---|---|---|---|---|---|
| 1 | 1.9 | 80.7 | 8.0 | 9.1 | - | 0.3 | - |
| 2 | 1.8 | 78.8 | 9.3 | 10.1 | - | - | - |
| 3 | 5.0 | 77.6 | 8.0 | 5.6 | 3.8 | - | - |
| 4 | 1.8 | 90.8 | 5.2 | 1.4 | 0.2 | - | 0.6 |
| 5 | 1.9 | 98.1 | - | - | - | - | - |

The AA3003 alloy exhibits a heterogeneous structure with numerous intermetallic particles (IMPs) with sizes ranging from nano to micrometers. Davoodi et al. [24,25] reported that larger (>2 μm) particles of the EN AW-3003 alloy are constituted of Al$_6$(Mn, Fe) type and also Al$_{12}$(Mn, Fe)$_3$Si type, with the Mn/Fe ratio being around 1:1 for both cases. This alloy contains numerous fine particles appearing as white dots, known as disperoids, sized about 0.5 μm; their composition corresponds to Al$_{12}$Mn$_3$Si$_{1–2}$. Similar results were found in this study.

On the backscattered SEM image (Figure 3) the IMPs were seen as brighter due to their larger atomic mass in comparison to the Al matrix. The larger micrometer-sized (about 3 μm) IMP at Location 1 (Figure 3, Table 2) contained Al(Mn, Fe) with a small amount of Cu. The ratio between Mn and Fe was about 0.9, which is consistent with the literature [24,25]. At Location 2, Al(Mn, Fe) IMP was observed with the Mn/Fe ratio of 0.9 as well. The IMP at Location 3 in comparison with IMPs 1 and 2 additionally contained 3.8 at.% Si, which typically denotes the Al(Mn, Fe)Si phase with a Mn/Fe ratio of about 1.4 and a Mn/Si ratio of 2.1. This particle was smaller (~1.3 μm) than IMPs 1 and 2. In addition to Al(Mn, Fe)Si, a small concentration of fluorine was present in a 2 μm large white spherical particle at Location 4. Fluorine originates from the alkaline cleaning agent. The ratio between Mn and Fe was larger (~3.6) than for IMP 3. Therefore, the differences between the IMPs 3 and 4 were the presence of F, a larger Mn/Fe ratio, and a much larger Mn/Si ratio (2.1 for IMP 3 and 26 for IMP 4). The composition at Location 5 corresponds to that of the matrix alloy surface containing Al (Table 2).

### 3.2. Conversion TCP Coatings on AA3003

Following the chemical pre-treatment (Figure A1), the samples were coated by immersion in a commercial conversion bath coating SurTec® 650 (denoted as ST650). The first step was to investigate the effect of the concentration of ST650 (10, 25 and 50 vol.%) at room temperature by keeping the conversion time constant, aiming to determine the appropriate concentration for sample preparation. The latter was then used in the second step to determine the appropriate conversion time (Figure 4). During coating formation in the conversion bath, the OCP of AA3003 was measured. The morphologies, compositions and corrosion properties of differently prepared TCP coatings were then analyzed.

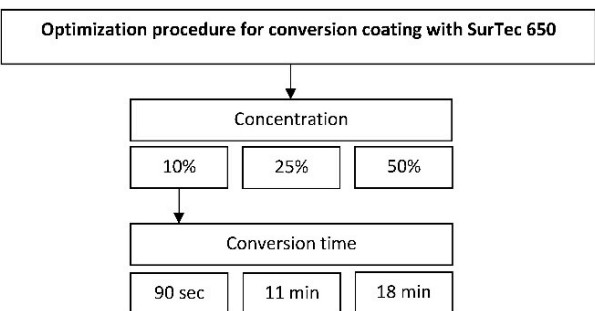

**Figure 4.** A schematic presentation of the selection of deposition conditions of conversion coating SurTec® 650 on the AA3003 substrate.

### 3.2.1. Effect of Concentration of the Properties of SurTec® 650 TCP

Electrochemical Measurements

Figure 5 compares the OCP curves recorded during the immersion of AA3003 in conversion baths of different concentrations of ST650.

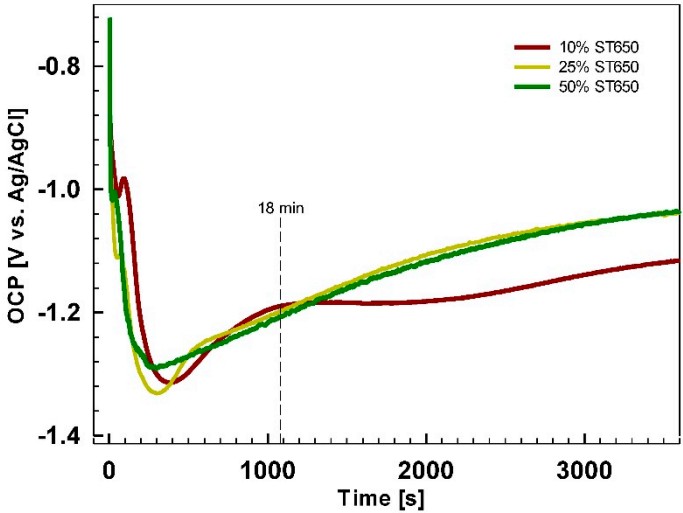

**Figure 5.** The dependence of the open circuit potential vs. time during the immersion of AA3003 in 10%, 25% and 50 vol.% SurTec 650 conversion bath at room temperature.

For all samples, OCP starts between −0.7 V and −0.8 V. The potential initially quickly decayed up to a small peak at values between −1.01 to −1.1 V. This peak was followed by a larger decay to minimum values at *cca.* 290 s for 50% ST650 (−1.29 V), *cca.* 300 s for 25% ST650 (−1.33 V) and *cca.* 360 s for 10% ST650 (−1.31 V). Subsequently, OCP curves gradually shifted to more positive values, attaining a relatively constant value. The latter was about 80 mV more positive for more concentrated TCP solutions. The initial decay of potential to a minimum of around −1.3 V was a consequence of the

dissolution of the passive Al oxide formed on the pre-treated surface by aggressive hexafluoride ions according to Reaction (3) [4]:

$$Al^{3+} + ZrF_6^{2-} \rightarrow AlF_6^{3-} + Zr^{4+} \tag{3}$$

This process corresponds to surface activation and represents the initiation stage of film formation [4,6,26]. The reactions of oxygen reduction and hydrogen evolution take place at cathodic sites:

$$O_2 + 2H_2O + 4e^- \rightarrow 4OH^- \tag{4}$$

$$2H^+ + 2e^- \rightarrow H_2 \tag{5}$$

Local alkalization due to Reactions (4) and (5) up to a pH of 8.5 establishes the conditions required for the deposition of Zr-hydroxide, which is a pH-driven process [4,27] along with the formation of Cr-hydroxide:

$$ZrF_6^{2-} \text{ (aq)} + 4OH^- \rightarrow ZrO_2 \cdot 2H_2O \text{ (s)} + 6F^- \text{ (aq)} \tag{6}$$

$$Cr^{3+} + 3OH^- \rightarrow Cr(OH)_3 \text{ (s)} \tag{7}$$

The precipitation and lateral growth of the conversion layer is reached at the minimum of the potential vs. time curve (*cca.* −1.3 V) where the rate of conversion coating precipitation starts to prevail over that of metal dissolution. A plateau reached at longer conversion times denotes the steady state deposition process, accompanied with extensive coverage of conversion coating. Further deposition is expected to lead to an increase in layer thickness without a change in potential [4,6]. For the samples formed at 10% ST650, the plateau was reached at *cca.* 18 min, whereas at larger concentrations, the potential was still shifting slightly more positive. The conversion time of 18 min was taken as a constant parameter when further studying the effect of ST650 concentration.

Typical polarization curves for the AA3003 sample coated with ST650 conversion coatings prepared at a conversion time of 18 min and at different concentrations of the conversion bath are shown in Figures 6 and 7. Electrochemical parameters deduced from these curves are given in Table 3 to compare the corrosion behavior for the different solution-treated samples.

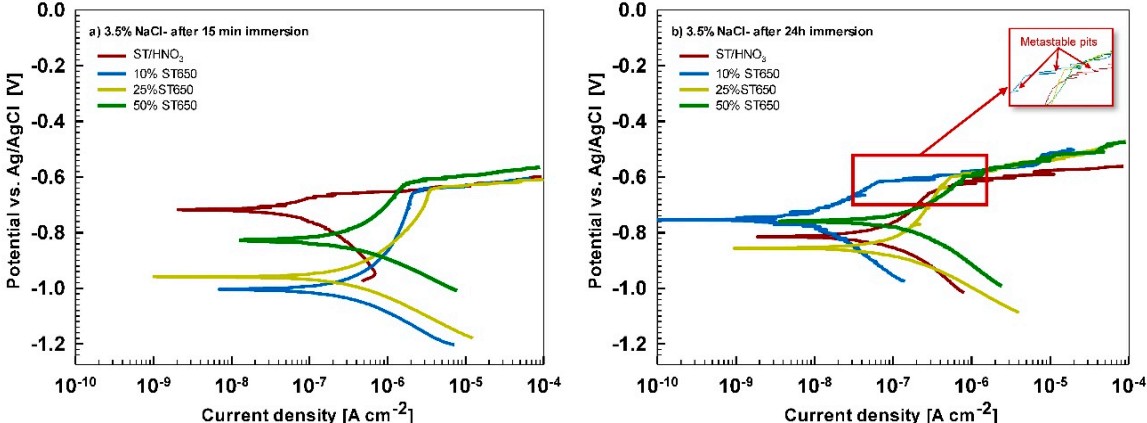

**Figure 6.** Potentiodynamic polarization curves recorded for uncoated, chemically pre-treated AA3003 (ST/HNO$_3$) and coated with ST650 coatings of different concentrations at a conversion time of 18 min at room temperature. The curves were recorded in 3.5% NaCl solution: (**a**) after 15 min immersion; (**b**) after 24 h immersion. The scan rate was 1 mV/s.

Figure 6a presents potentiodynamic polarization curves measured in 3.5% NaCl solution after 15 min immersion of chemically pre-treated AA3003 and coated with ST650 of different concentrations. The presence of 10% and 25% ST650 coatings caused a 210 mV and 170 mV shift of $E_{corr}$, respectively, in the more cathodic direction and somewhat less, 130 mV, for 50% ST650 (Table 3). This may indicate that the TCP coating provided better cathodic than anodic inhibition, as also indicated by the broadening of

$\Delta E$. However, for the chemically pre-treated sample, the $j_{corr}$ value was lower than for coated samples. All samples had similar $E_{bd}$. These data indicate that after 15 min of immersion, the coated samples did not act protectively.

Changes in the shape of polarization curves were noticeable after 24 h immersion in 3.5% NaCl solution (Figure 6b). Compared with the results after 15 min immersion, all curves for coated samples shifted in the positive direction and towards smaller current densities. All samples exhibited similar $E_{corr}$ (between −0.76 V and −0.84 V). However, the best parameters—largest $R_p$ and smallest $j_{corr}$—were observed for the 10% ST650-coated samples. Furthermore, this coating showed reduced anodic and cathodic currents in the whole potential range, indicating that the coating affected both processes. The reduction of the anodic current is a consequence of passivation of the Al matrix by the conversion coating in the surrounding area of the intermetallic parts [7]. The reduction of oxygen takes place in the cathodic current region. This conversion coating thus provides a physical barrier to migration of dissolved oxygen to the surface. In contrast, the coatings of other two concentrations (25% and 50% ST650) showed similar behaviors as the untreated sample. The curves indicate the occurrence of some metastable pits, as shown in the inset in Figure 6b. The pitting attack could happen due to the presence of Cl$^-$ ions at the interface between silicon particles and the $\alpha$-Al phase. Silicon is cathodic with respect to the Al-rich matrix, which may lead to the formation of micro-galvanic couples and generate the pitting corrosion [28].

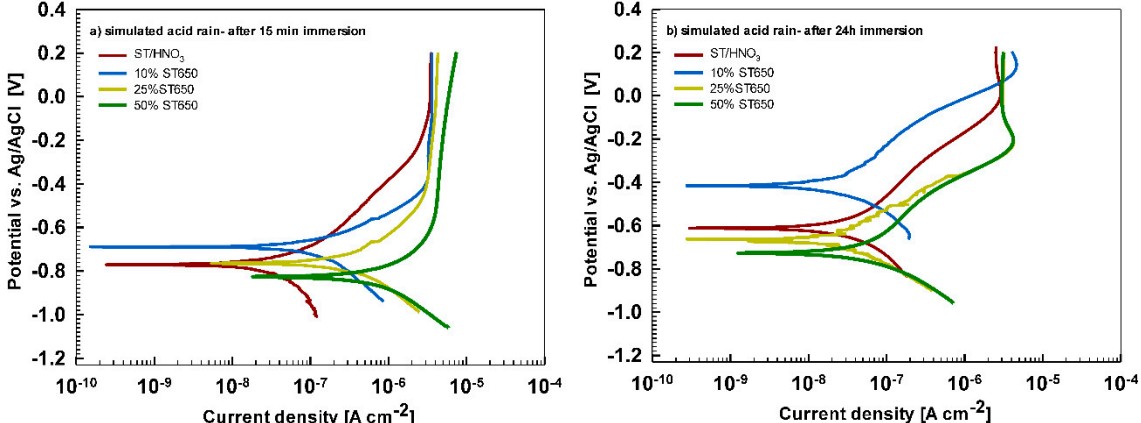

**Figure 7.** Potentiodynamic polarization curves recorded for uncoated, chemically pre-treated AA3003 (ST/HNO$_3$) and coated with ST650 coatings of different concentrations at a conversion time of 18 min at room temperature. The curves were recorded in simulated acid rain solution: (**a**) after 15 min immersion; (**b**) after 24 h immersion. The scan rate was 1 mV/s.

**Table 3.** Electrochemical parameters deduced from potentiodynamic measurements in 3.5% NaCl and simulated acid rain solution for chemically pre-treated AA3003 (ST/HNO$_3$) coated with different concentrations of ST650. The polarization resistance ($R_p$) was determined from linear the polarization measurements, and the corrosion potential ($E_{corr}$), corrosion current density ($j_{corr}$), breakdown potential ($E_{bd}$) and $\Delta E = |E_{bd} - E_{corr}|$ from potentiodynamic polarization curves (Figures 6 and 7). Values are given as mean ± standard deviation.

| Electrolyte | Immersion Time | Sample | $R_p$/k$\Omega$ cm$^2$ | $E_{corr}$/V | $j_{corr}$/$\mu$A cm$^{-2}$ | $E_{bd}$/V | $\Delta E$/V |
|---|---|---|---|---|---|---|---|
| 3.5% NaCl | 15 min | ST/HNO$_3$ | 682 ± 85 | −0.74 ± 0.03 | 0.04 ± 0.003 | −0.67 ± 0.01 | 0.07 |
| | | 10% ST650 | 187 ± 96 | −0.95 ± 0.07 | 0.27 ± 0.09 | −0.65 ± 0.002 | 0.30 |
| | | 25% ST650 | 129 ± 54 | −0.91 ± 0.07 | 0.29 ± 0.05 | −0.64 ± 0.01 | 0.26 |
| | | 50% ST650 | 75 ± 7 | −0.87 ± 0.06 | 0.30 ± 0.09 | −0.63 ± 0.003 | 0.24 |
| | 24 h | ST/HNO$_3$ | 735 ± 299 | −0.80 ± 0.02 | 0.06 ± 0.03 | −0.64 ± 0.02 | 0.17 |
| | | 10% ST650 | 1927 ± 73 | −0.76 ± 0.01 | 0.01 ± 0.01 | −0.59 ± 0.04 | 0.17 |
| | | 25% ST650 | 573 ± 268 | −0.84 ± 0.02 | 0.07 ± 0.03 | −0.61 ± 0.01 | 0.23 |
| | | 50% ST650 | 459 ± 256 | −0.79 ± 0.04 | 0.10 ± 0.05 | −0.61 ± 0.02 | 0.18 |

<div align="center"><b>Table 3.</b> <i>Cont.</i></div>

| Electrolyte | Immersion Time | Sample | $R_p$/kΩ cm$^2$ | $E_{corr}$/V | $j_{corr}$/μA cm$^{-2}$ | $E_{bd}$/V | $\Delta E$/V |
|---|---|---|---|---|---|---|---|
| Simulated acid rain | 15 min | ST/HNO$_3$ | 825 ± 229 | −0.76 ± 0.01 | 0.06 ± 0.03 | - | - |
| | | 10% ST650 | 348 ± 122 | −0.64 ± 0.07 | 0.23 ± 0.13 | - | - |
| | | 25% ST650 | 155 ± 9 | −0.83 ± 0.1 | 0.44 ± 0.09 | - | - |
| | | 50% ST650 | 73 ± 11 | −0.84 ± 0.03 | 0.80 ± 0.06 | - | - |
| | 24 h | ST/HNO$_3$ | 1974 ± 793 | −0.59 ± 0.10 | 0.03 ± 0.01 | - | - |
| | | 10% ST650 | 1176 ± 462 | −0.57 ± 0.22 | 0.03 ± 0.02 | - | - |
| | | 25% ST650 | 1027 ± 97 | −0.66 ± 0.01 | 0.03 ± 0.01 | - | - |
| | | 50% ST650 | 1232 ± 559 | −0.67 ± 0.07 | 0.04 ± 0.02 | - | - |

Potentiodynamic polarization curves for chemically pre-treated AA3003 coated with ST650 coatings of different concentrations were also studied in simulated acid rain solution. Figure 7a reveals that $R_p$ decreased and $j_{corr}$ increased for more concentrated coatings. The $E_{corr}$ for coated sample 10% ST650 shifted more positively. At potentials above −0.08 V, similar anodic current densities were attained for the coated and uncoated samples. The cathodic current densities for the coated samples increased compared to the untreated sample. These results indicate that after short immersion these conversion coatings were not efficient. However, the behavior changed for longer immersion, similarly to the chloride solution (Figure 6b). After 24 h in simulated acid rain (Figure 7b), $E_{corr}$ shifted to more positive values, the most positive being achieved for 10% ST650 (Table 3). The corrosion current density at *cca.* 0.03 μA cm$^{-2}$, determined from Tafel extrapolation, was quite similar for all conversion-coated samples (Table 3).

SEM/EDS and XPS Characterization

SEM, EDS and XPS characterizations were performed to investigate the morphology and chemical composition of the coatings of different ST650 concentrations.

The SEM micrographs presented in Figure 8 show that at 10% ST650, the surface was generally similar to the untreated sample, but the presence of a surface layer can be noted (Figure 8a). When prepared from more concentrated ST650 solutions, coatings were micro-cracked (Figure 8b,c) which may be related to dehydration in vacuum and/or drying process of the coating. Although cracking does not necessarily imply bad corrosion protection [7], the fact is that more concentrated coatings exhibited worse corrosion resistance (Figures 6 and 7).

EDS analyses at different locations revealed that elements Zr, Cr and F, belonging to conversion coatings, were detected for samples prepared at all three concentrations (Table 4). The highest ratio Zr/Cr was obtained for the 10% ST650 sample, i.e., between 1.3 and 3.7. The IMP Al(Mn, Fe) at Location 1 in addition to Zr and Cr also contained fluorine and a small amount of sulfur. The ratio between Mn and Fe was around 1.4. At Location 2, a similar particle was observed as IMP 1, but without F and S. The Mn/Fe ratio was about 0.9. IMP 3, a 0.5 μm sized particle Al(Mn, Fe)Si, contained 0.4 at.% Cu and 1.9 at.% F. A Mn/Fe ratio of about 1.1 and Mn/Si ratio of 1.4 was calculated. The highest ratio of Zr/Cr was obtained at this Location (~3.7). IMP 4 also belongs to the Al(Mn, Fe)Si phase as IMP 3, but without Cu and S. The Mn/Fe ratio was 1.4 and Mn/Si 1.8.

The coating formed using 25% ST650 contained *cca.* one micrometer-sized white spherical Al(Mn, Fe) IMP at Location 5 with Zr, Cr, F and O (Figure 8b, Table 4). The ratio Mn/Fe was about 1.4. At Location 6, the Al(Mn, Fe)Si phase with an Mn/Fe ratio of about 1.3 and Mn/Si ratio of 1.7 was detected. Rod-shaped Al(Mn, Fe) IMP 7 in addition to Zr, Cr and O also contained 0.2 at.% Cu. Fluorine was not present. The Mn/Fe ratio of 0.8 was lower than for IMP 5. IMP 8 was smaller than 0.5 μm and contained Al, Mn, O, Cr, Zr and F. No Fe was detected. The ratio Zr/Cr at 1.5 remained the same at all locations.

Conversion ST650 coating of the highest concentration contained numerous micro cracks, as shown in Location 9, where only O, Al and a small amount of Mn were present (Figure 8c, Table 4). No elements originating from the TCP coating were detected. The IMP Al(Mn, Fe)Si at Location 10

contained Zr/Cr in a ratio of 1:1 and in addition to F and O, sulfur was also present. The ratio Mn/Fe and Mn/Si were both 1.3. Rod-shaped grey colored particles of Al(Mn, Fe) at Location 11 were similar in composition to IMP 7, with a small amount of Cu. However, the main difference between IMPs 7 and 11 was in a higher Mn/Fe ratio in IMP 11 (~2.5). At Location 12, one micrometer-sized IMP corresponded to Al(Mn,Fe)Si, with a somewhat smaller concentration of Zr and Cr. Mn/Fe and Mn/Si ratios were 1.1 and 1.4, respectively. The Zr/Cr ratio on the 50% ST650-coated samples was between 1 and 1.3 (Table 4).

Compared to more concentrated coatings, the 10% ST650 coating contained less O and more Al, Mn and Fe. This may be related to a discrepancy between the relatively large analysis depth of EDS and the small coating thickness, which was around 100 nm [7]. As will be shown below, XPS is a more appropriate technique for such thin coatings.

The high-magnification SEM images of the coating surfaces prepared with different concentrations of ST650 are compared in Figure 9. The 10% ST650 coating shows a uniform morphology consisting of nodules about 100 nm in diameter (Figure 9a). The nodules on the 25% ST650 coated surface were larger (Figure 9b), up to 180 nm. Cracks were observed, as has been previously reported [6,7,29–31] and ascribed to the cracks that formed during coating growth and/or subsequent drying in vacuum. Figure 9c corresponds to the conversion coating with 50% ST650. The particles of diameters around 75 nm were agglomerated non-uniformly on the surface.

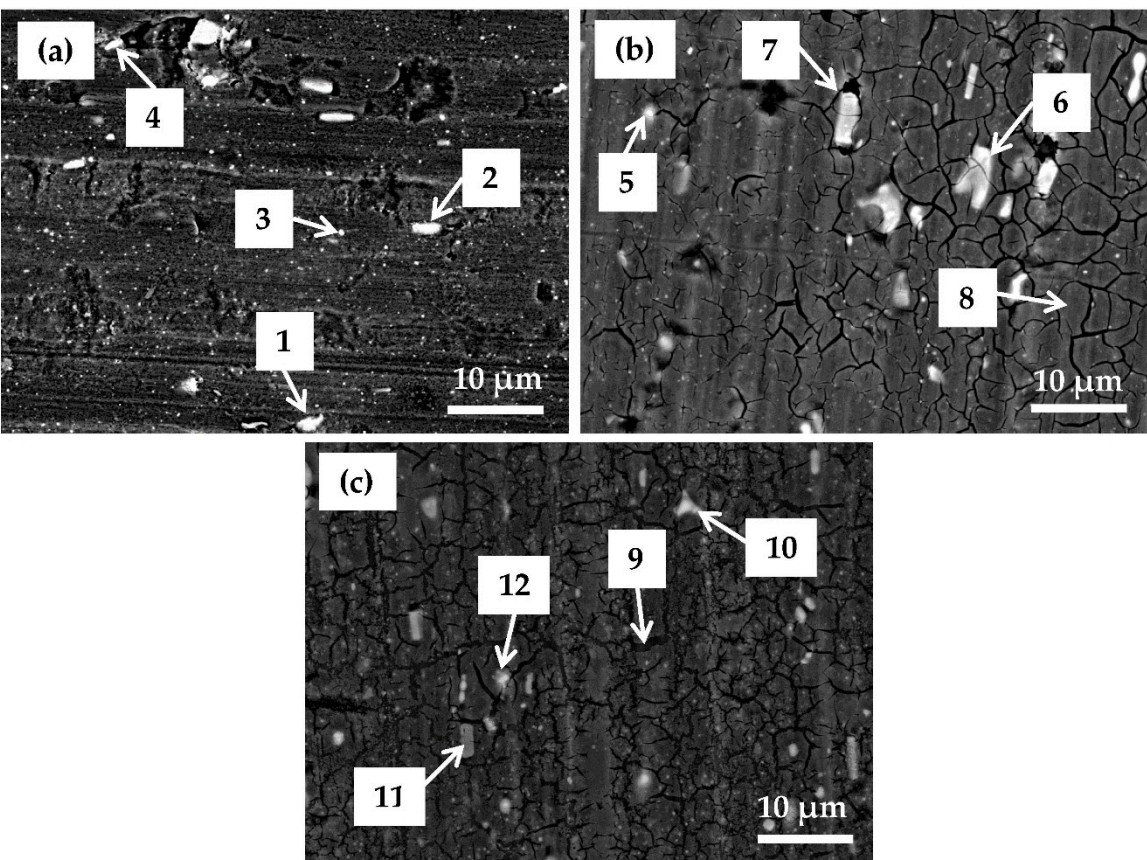

**Figure 8.** SEM backscattered electron micrographs for conversion coated AA3003 with different ST650 concentrations: (**a**) 10% ST650; (**b**) 25% ST650 and (**c**) 50% ST650. The conversion time was 18 min at room temperature. The enumerated spots are the locations at which EDS analysis was performed (Table 4).

**Table 4.** The concentrations of elements obtained by EDS analysis of AA3003 coated with different concentrations of ST650 (Figure 8).

| Location | Sample | at.% | | | | | | | | | |
|---|---|---|---|---|---|---|---|---|---|---|---|
| | | O | Al | Mn | Fe | Si | Cr | Zr | F | Cu | S |
| 1 | | 10.9 | 70.6 | 8.1 | 5.6 | - | 1.2 | 1.6 | 1.7 | - | 0.2 |
| 2 | 10% | 11.2 | 72.3 | 6.4 | 7 | - | 1.2 | 1.9 | - | - | - |
| 3 | ST650 | 8.4 | 73.2 | 5.6 | 5.1 | 4 | 0.3 | 1.1 | 1.9 | 0.4 | - |
| 4 | | 6.5 | 69.7 | 10 | 7.2 | 5.7 | 0.3 | 0.7 | - | - | - |
| 5 | | 57.8 | 35 | 2.6 | 1.8 | - | 0.2 | 0.3 | 2.4 | - | - |
| 6 | 25% | 58.5 | 32.7 | 2.9 | 2.2 | 1.7 | 0.3 | 0.3 | 1.5 | - | - |
| 7 | ST650 | 59.2 | 36.1 | 1.8 | 2.2 | - | 0.2 | 0.3 | - | 0.2 | - |
| 8 | | 59.2 | 38.9 | 0.1 | - | - | 0.2 | 0.3 | 1.4 | - | - |
| 9 | | 60.0 | 39.9 | 0.1 | - | - | - | - | - | - | - |
| 10 | 50% | 57.2 | 28.8 | 2.5 | 1.9 | 1.9 | 1.5 | 1.5 | 4.6 | - | 0.2 |
| 11 | ST650 | 59 | 37.5 | 1 | 0.4 | - | 0.3 | 0.4 | 1.3 | 0.1 | - |
| 12 | | 59.5 | 31.9 | 2.8 | 2.5 | 2 | 0.5 | 0.6 | - | 0.1 | 0.1 |

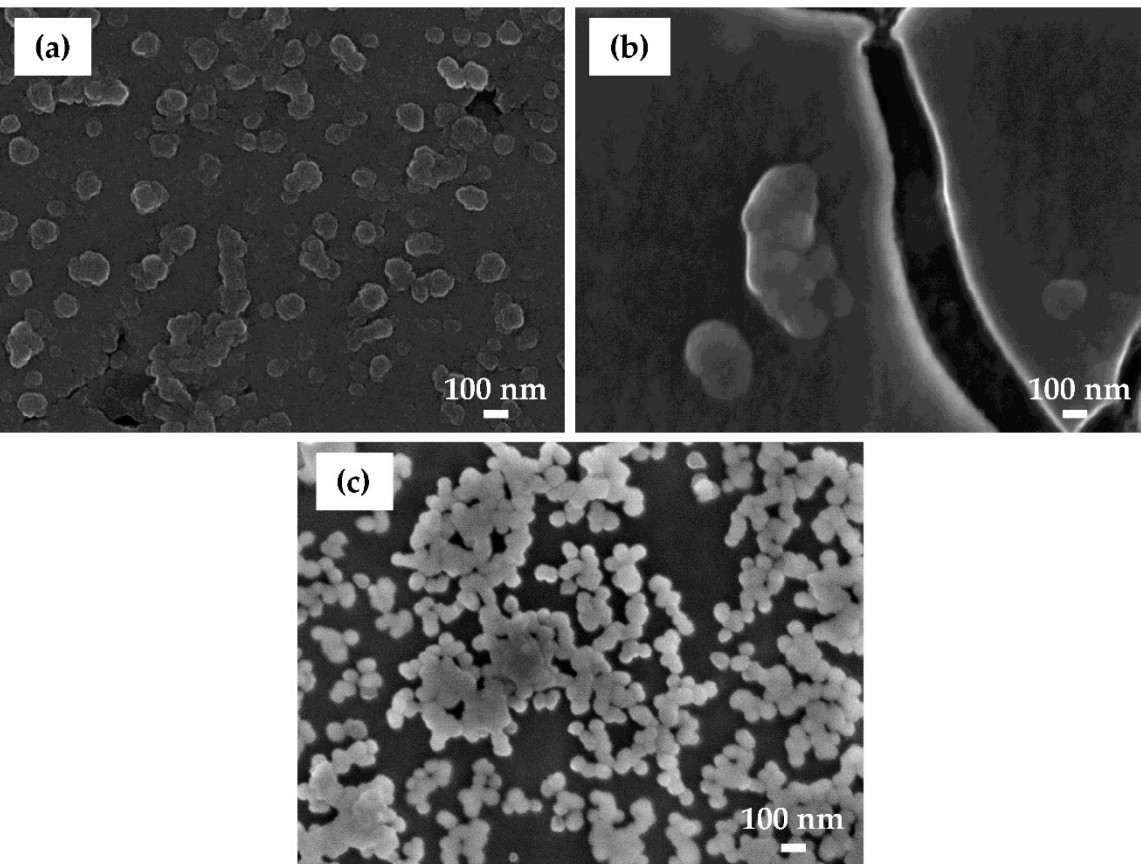

**Figure 9.** SEM electron secondary electrons micrographs for conversion-coated AA3003 with ST650 coatings of different concentrations: (**a**) 10% ST650; (**b**) 25% ST650 and (**c**) 50% ST650. The conversion time was 18 min at room temperature. Magnification 50,000×.

XPS analysis was also carried out on ST650 coatings of different concentrations. Compared to the EDS analysis, the analyzed spot of XPS was laterally larger but with a smaller depth of analysis, thus giving a response from the near-surface region of up to 10 nm. Therefore, EDS analysis serves more as an orientating method once the surface is covered by a conversion layer. In addition, XPS analysis enables the speciation of the oxidation states of elements. The surface elemental concentration is shown in Table 5. The surface contained O, Al, Zr, Cr, F and P. The oxygen content was high, indicating an abundant oxide/hydroxide formation. The latter can be related to Al-, Zr-, and Cr-hydrated oxides. Fluorine originates from the conversion bath and phosphorus may be related to the chemical pre-treatment. At the lowest concentration of the conversion bath (10% ST650), the largest Zr and F contents (Table 5) were noted, along with the smallest Al content, which is in discrepancy with EDS results (Table 4). As XPS is related to the near-surface region. This may indicate that the coating composition changes with coating depth, i.e., it exhibits a bi-layer or even tri-layer structure, as pointed out previously [7–10].

**Table 5.** The concentrations of elements obtained by XPS analysis for the AA3003 samples coated with ST650 conversion coatings of different concentrations: 10%, 25% and 50%.

| Sample | at.% O | at.% Al | at.% Cr | at.% Zr | at.% F | at.% P |
|--------|--------|---------|---------|---------|--------|--------|
| 10% ST650 | 66.4 | 5.6 | 1.8 | 16.3 | 8.4 | 1.5 |
| 25% ST650 | 77.3 | 13.9 | 4.2 | 3.9 | 0.6 | - |
| 50% ST650 | 76.5 | 13.2 | 2.7 | 4.2 | 1 | 2.4 |

At 10% ST650, Zr- and Cr-oxides and fluorides were readily formed, resulting in the formation of a uniform layer covering the underlying substrate, as shown by SEM (Figure 9a). With the increasing bath concentration, the Zr/Cr ratio decreased from ~9 to 0.9, being the lowest for 25% ST650. At the same time, the content of O prevailed largely over that of F. The increased Al concentration, observed at more concentrated coatings (Table 5), may be related to the underlying substrate which can be detected due to the presence of cracks within the coating, or Al may have been partially incorporated in the coating. Regardless of the concentration of ST650, Al was present predominantly as Al hydroxide (Figure 10, fitted spectra not shown). With the increasing concentration of the conversion bath, the contents of Zr, Cr and F decreased, which may be related to a reduced oxide/hydroxide formation and/or a more non-uniform coating, as shown in Figure 9b,c.

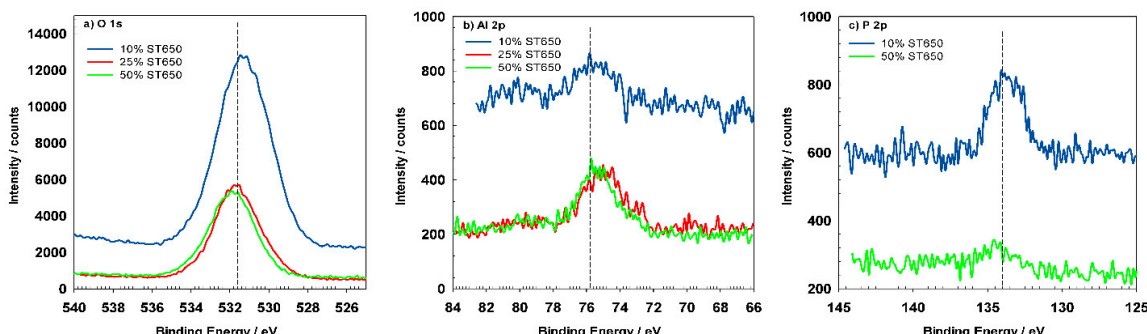

**Figure 10.** High-resolution XPS spectra of (**a**) O *1s*; (**b**) Al *2p*; (**c**) P *2p* recorded for AA3003 samples coated with ST650 conversion coatings of different concentrations: 10%, 25% and 50%. The conversion time was 18 min at room temperature. Spectra were not normalized on the intensity scale.

Spectra for O, Al, P, Cr, Zr and F, as main constituent of surface conversion coating ST650, were studied in more detail using high resolution XPS spectra (Figures 10 and 11). Figure 10a presents the high-resolution spectra of O *1s* photoelectron regions. The peaks are centered at the binding energy ($E_b$) 531.6 eV, as expected for a hydrated oxide. The Al *2p* spectra in Figure 10b are centered at a $E_b$ of 75.8 eV and are related to $Al_2O_3$ and $AlO_xF$ [6]. The P *2p* peak centered at 134 eV was only detected

for 10% and 50% ST650 coatings (Figure 10c). It was related to phosphate and originated from the SurTec cleaner.

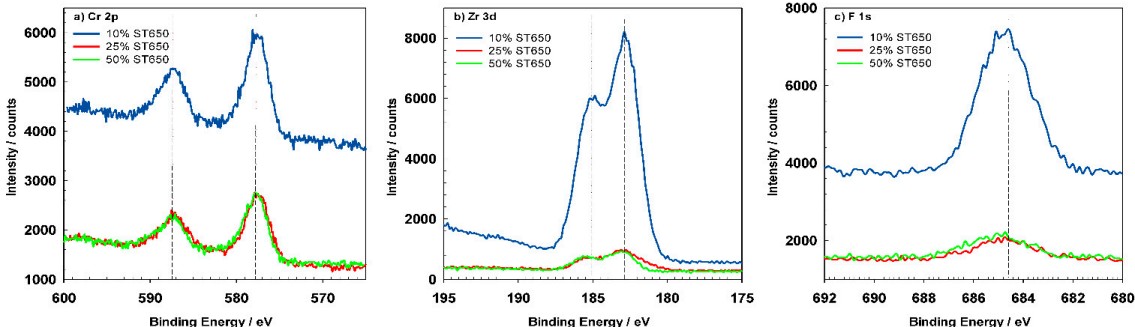

**Figure 11.** High-resolution XPS spectra of (**a**) Cr *2p*; (**b**) Zr *3d*; (**c**) F *1s* recorded for AA3003 samples coated with ST650 conversion coatings of different concentrations: 10%, 25% and 50%. The conversion time was 18 min at room temperature. Spectra were not normalized on the intensity scale.

Figure 11a shows the Cr *2p* spectra with Cr $2p_{3/2}$ and $2p_{1/2}$ peaks centered at 577.8 eV and 587.4 eV. The position of these peaks confirms the formation of hydrated Cr(III) oxide, as will be shown by spectra deconvolution, taking into account component sub-peaks for Cr metal, $Cr_2O_3$, $Cr(OH)_3$, Cr(VI) oxide and $CrF_3$. The Zr *3d* region (Figure 11b) presents two double split orbits separated by *cca.* 2 eV related to Zr $3d_{5/2}$ and $3d_{3/2}$ peaks. These peaks can be sub-divided into $ZrO_2$ and $ZrF_4$ (see below). The intensity of Zr *3d* peaks was much higher for 10% ST650 than for more concentrated coatings. Similarly, the peak of F (Figure 11c) had a higher intensity for the sample with 10% ST650 compared to 25% and 50% ST650. The presence of F proves that fluorine was incorporated in the coating, especially at the lowest concentration of the conversion bath.

### 3.2.2. Effect of Conversion Time of the Properties of SurTec® 650 TCP

Based on the results obtained in the previous section, the following parameters of the conversion process were used to study the effect of conversion time: 10 vol.% SurTec® 650 and conversion times 90 s, 11 min and 18 min at room temperature. These values were selected based on the OCP vs. time curve (Figure 5) and reflect three stages of coating formation: initial activation (90 s), precipitation (11 min) and growth (18 min).

Electrochemical Measurements

Potentiodynamic polarization curves were recorded in 3.5% NaCl and simulated acid rain after 15 min and 24 h immersion time in order to follow the behavior of coated samples as a function of time (Figures 12 and 13, Table 6). The polarization curve recorded for chemically pre-treated sample $ST/HNO_3$ is given for the sake of comparison.

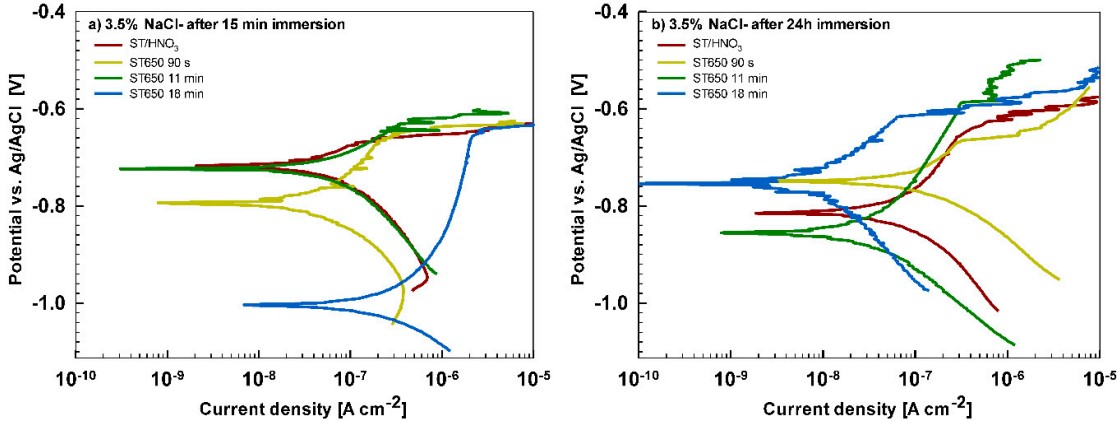

**Figure 12.** Potentiodynamic polarization curves in 3.5% NaCl after (**a**) 15 min immersion; (**b**) 24 h immersion for AA3003 samples coated with 10% ST650 prepared for different conversion times: 90 s, 11 min and 18 min at room temperature. The samples were chemically pre-treated using ST/HNO$_3$ and the bare sample is given for comparison. The scan rate was 1 mV/s.

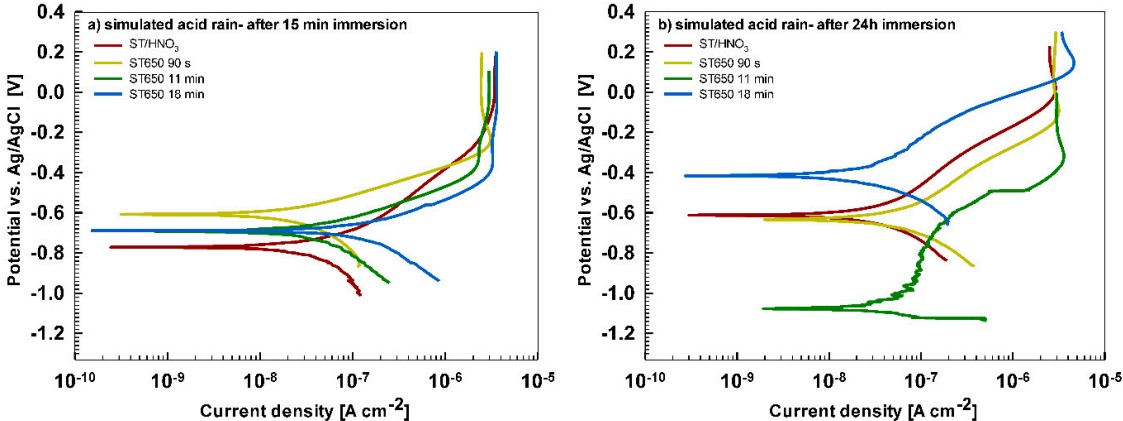

**Figure 13.** Potentiodynamic polarization curves in simulated acid rain after (**a**) 15 min immersion; (**b**) 24 h immersion for AA3003 samples coated with 10% ST650 prepared for different conversion times: 90 s, 11 min and 18 min at room temperature. The samples were chemically pre-treated using ST/HNO$_3$ and the bare sample is given for comparison. The scan rate was 1 mV/s.

After 15 min stabilization in 3.5% NaCl, the curves recorded for coated AA3003 samples differed depending on the conversion time (Figure 12a). The coating prepared for 11 min was similar to that of the uncoated sample; the other two curves were shifted in a negative direction. None of the curves showed an improvement in corrosion parameters compared to the bare sample (Table 6). However, with prolonged immersion in NaCl (from 15 min to 24 h, Figure 12b) the behavior of uncoated and, especially, coated samples changed. The uncoated sample exhibited a slight improvement in corrosion properties, probably related to the development of the naturally grown oxide layer able to temporarily resist the attack of chloride ions (Figure 1b). The coated sample prepared at a 90 s conversion time showed degraded properties after 24 h immersion in NaCl compared to 15-min immersion. This coating was thus not protective at longer immersion times. In contrast, coatings deposited at 11 min and 18 min showed considerably larger $R_p$ (4- to 10-fold compared to 15 min immersion) and a smaller $j_{corr}$ (15- to 27-fold compared to 15 min immersion). During prolonged immersion, a large positive shift of the curve for the sample deposited for 18 min occurred changing $E_{corr}$ from −0.95 V to −0.76 V. The sample deposited for 11 min showed a shift in the opposite direction, i.e., from −0.74 V to −0.86 V. This also affected $E_{bd}$ values. After 15 min in NaCl, the $E_{bd}$ values were very similar to the chemically treated sample. After 24 h in NaCl, however, the coatings deposited for 11 and 18 min showed a shift in $E_{bd}$ for 90 mV and 50 mV more positive compared to the bare alloy, respectively. Therefore,

at an equivalent conversion time in the ST650 bath (11 and 18 min), the corrosion parameters of the coated samples were improved after prolonged immersion in NaCl (smaller current density, larger polarization resistance and a more positive breakdown potential). This result indicates that with a longer immersion time, the corrosion resistance increased, resulting in a better protectivity of the layer to the localized breakdown.

Substrate pre-treatment beneficially influences the subsequent coating process. The comparison of ST650-coated AA3003 substrates prepared with and without chemical pretreatment with ST/HNO$_3$ is presented in Figure A6 and Table A2. Potentiodynamic curves were recorded after 15 min and 24 h immersion in 3.5 wt.% NaCl. It is evident that a better corrosion resistance was achieved on chemically pre-treated substrate.

In less aggressive electrolyte of simulated acid rain, the initial attack was less severe than in 3.5% NaCl (Figure 13). After 15-min immersion (Figure 13a) the coating deposited for 90 s showed the most favorable corrosion parameters whilst the coating deposited at 18 min was the least resistant (Table 6). After 24 h immersion (Figure 13b), however, a similar behavior as in NaCl solution was observed: the curve for the coating deposited for 18 min shifted strongly in the positive direction and exhibited the most favorable corrosion parameters (smallest current density, largest polarization resistance and most positive corrosion potential) (Table 6).

**Table 6.** The polarization resistance ($R_p$), corrosion potential ($E_{corr}$), corrosion current density ($j_{corr}$), breakdown potential ($E_{bd}$) and the extent of the passive range $\Delta E = |E_{bd} - E_{corr}|$ measured in 3.5% NaCl and simulated acid rain after 15 min and 24 h immersion for chemically pre-treated AA3003 (ST/HNO$_3$) and coated AA3003 specimens after various conversion times in ST650: 90 s, 11 min and 18 min. Values are given as mean ± standard deviation.

| Electrolyte | Immersion Time | Sample | $R_p$/k$\Omega$ cm$^2$ | $E_{corr}$/V | $j_{corr}$/µA cm$^{-2}$ | $E_{bd}$/V | $\Delta E$/V |
|---|---|---|---|---|---|---|---|
| 3.5% NaCl | 15 min | ST/HNO$_3$ | 682 ± 85 | −0.74 ± 0.03 | 0.04 ± 0.003 | −0.67 ± 0.01 | 0.07 |
| | | ST650 90 s | 452 ± 22 | −0.75 ± 0.06 | 0.05 ± 0.01 | −0.64 ± 0.01 | 0.11 |
| | | ST650 11 min | 417 ± 341 | −0.74 ± 0.03 | 0.47 ± 0.44 | −0.63 ± 0.01 | 0.11 |
| | | ST650 18 min | 187 ± 96 | −0.95 ± 0.07 | 0.27 ± 0.09 | −0.65 ± 0.002 | 0.30 |
| | 24 h | ST/HNO$_3$ | 735 ± 299 | −0.80 ± 0.02 | 0.06 ± 0.03 | −0.64 ± 0.02 | 0.17 |
| | | ST650 90 s | 166 ± 39 | −0.77 ± 0.02 | 0.16 ± 0.06 | −0.51 ± 0.02 | 0.25 |
| | | ST650 11 min | 1144 ± 462 | −0.86 ± 0.001 | 0.03 ± 0.01 | −0.55 ± 0.05 | 0.30 |
| | | ST650 18 min | 1927 ± 73 | −0.76 ± 0.01 | 0.01 ± 0.01 | −0.59 ± 0.04 | 0.17 |
| Simulated acid rain | 15 min | ST/HNO$_3$ | 825 ± 229 | −0.76 ± 0.01 | 0.06 ± 0.03 | - | - |
| | | ST650 90 s | 1351 ± 1012 | −0.54 ± 0.10 | 0.06 ± 0.04 | - | - |
| | | ST650 11 min | 839 ± 394 | −0.70 ± 0.01 | 0.09 ± 0.06 | - | - |
| | | ST650 18 min | 348 ± 122 | −0.64 ± 0.07 | 0.23 ± 0.13 | - | - |
| | 24 h | ST/HNO$_3$ | 1974 ± 793 | −0.59 ± 0.10 | 0.03 ± 0.01 | - | - |
| | | ST650 90 s | 658 ± 195 | −0.67 ± 0.04 | 0.04 ± 0.04 | - | - |
| | | ST650 11 min | 575 ± 164 | −1.04 ± 0.05 | 0.05 ± 0.01 | −0.38 ± 0.16 | 0.66 |
| | | ST650 18 min | 1545 ± 716 | −0.66 ± 0.22 | 0.03 ± 0.02 | - | - |

SEM/EDS and XPS Characterization

Samples prepared at selected conversion times were analyzed using SEM/EDS to investigate the morphology and composition of the coating as a function of conversion time. Related SEM/EDS analysis is given in Figure 14 and Table 7.

After 90 s conversion (Figure 14a), the surface looked similar to the chemically pre-treated sample (Figure 3). Two types of white particles were detected on this coated surface. EDS microanalysis showed that one type was Al(Mn, Fe) IMPs (Location 1) and the second was Al(Mn, Fe)Si (Location 2). The content of O was rather low. No F or Zr were detected. At Location 3, only the matrix appeared. These results indicate that after 90 s immersion, the conversion coating did not form, or that its thickness was too small to be detected by EDS. It can be concluded that the potential peak observed after 90 s conversion at ~−1.0 V (Figure 5) was not related to coating formation but maybe to the oxidation of some chemical present in the bath. At Location 3, only Al was detected.

After conversion for 11 min (Figure 14b), a significant change in the surface composition took place: the contents of alloying elements Fe and Mn were smaller, the content of O was larger, and F, Cr and Zr were detected (Table 7). Increased O, F, Cr and Zr contents relate to the increased formation of oxides and fluorides. The coating was still not clearly visible in SEM. After conversion for 18 min (Figure 14c), however, the surface morphology changed; the IMPs were visible underlying the coating, indicating the formation of coating throughout the sample surface, i.e., at alloy matrix and at intermetallic particles. The concentrations of O, Cr and Zr were similar as after 11 min (Figure 14c, Table 7). IMP at Location 8 in comparison with other IMPs contained 0.2 at.% S. The highest ratio of Zr/Cr was obtained at Location 10 (~3.7). Elements Zr and Cr originating from the coatings were detected at the alloy matrix as well (Location 12). Therefore, coatings were deposited throughout the surface, but preferentially at intermetallic particles.

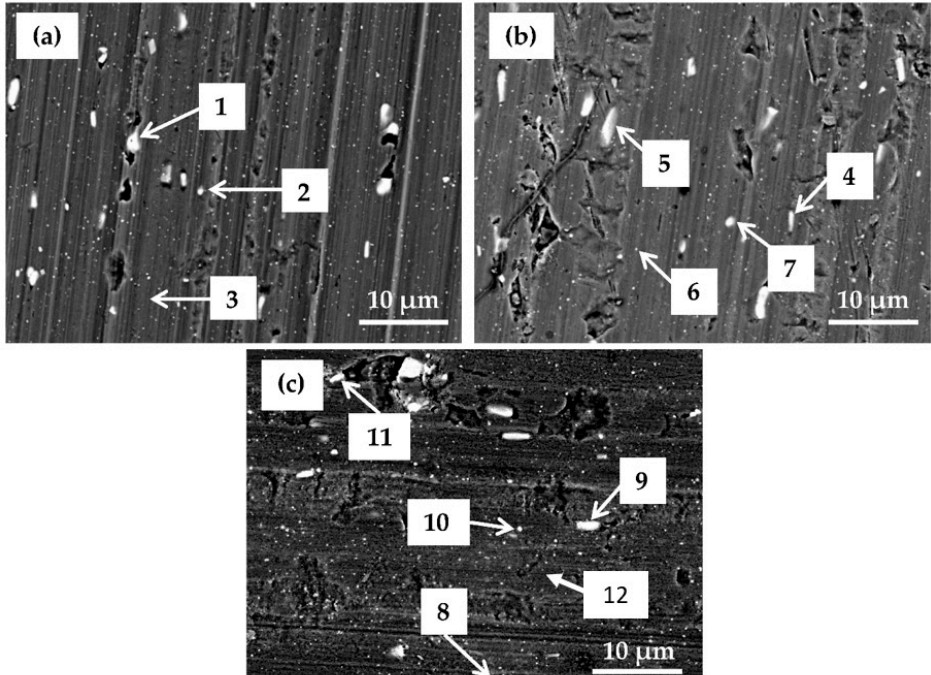

**Figure 14.** SEM backscattered electron micrographs for AA3003 coated with 10% ST650 conversion coatings prepared at different conversion times: (**a**) 90 s; (**b**) 11 min and (**c**) 18 min at room temperature. The enumerated spots are the locations at which EDS analysis was performed (Table 7).

**Table 7.** The concentrations of elements obtained by EDS analysis at the surface of 10% ST650 coating prepared at different conversion times (Figure 14); (**a**) 90 s (Locations 1–3); (**b**) 11 min (Locations 4–7) and (**c**) 18 min (Locations 8–11). Please note that the latter are the same as Locations 1–4 in Table 4.

| Location | Sample | at.% | | | | | | | | | |
| --- | --- | --- | --- | --- | --- | --- | --- | --- | --- | --- | --- |
| | | O | Al | Mn | Fe | Si | Cr | Zr | F | Cu | S |
| 1 | | 2.2 | 77.4 | 11.8 | 8.6 | - | - | - | - | - | - |
| 2 | 90 s | 3.1 | 70.9 | 11.3 | 9 | 5.8 | - | - | - | - | - |
| 3 | | 1.7 | 98.3 | - | - | - | - | - | - | - | - |
| 4 | | 13.6 | 68.7 | 7.5 | 4.7 | - | 0.9 | 1.9 | 2.9 | - | - |
| 5 | 11 | 6.2 | 77 | 7.8 | 8.6 | - | - | 0.5 | - | - | - |
| 6 | min | 5.6 | 86.9 | 2.7 | 1.2 | 1.5 | 0.3 | 0.7 | 1.2 | - | - |
| 7 | | 10.2 | 64.4 | 6.7 | 7.9 | 5.3 | 0.9 | 1.3 | 2.9 | 0.4 | - |
| 8 | | 10.9 | 70.6 | 8.1 | 5.6 | - | 1.2 | 1.6 | 1.7 | - | 0.2 |
| 9 | 18 | 11.2 | 72.3 | 6.4 | 7 | - | 1.2 | 1.9 | - | - | - |
| 10 | min | 8.4 | 73.2 | 5.6 | 5.1 | 4 | 0.3 | 1.1 | 1.9 | 0.4 | - |
| 11 | | 6.5 | 69.7 | 10 | 7.2 | 5.7 | 0.3 | 0.7 | - | - | - |
| 12 | - | 3.5 | 95.7 | - | - | - | 0.2 | 0.4 | - | - | 0.2 |

XPS spectra were recorded on chemically pre-treated AA3003 sample without the conversion coating and for ST650-coated AA3003 after various conversion times (Figures 15 and 16). The composition of the surface deduced from survey spectra is given in Table 8. The composition of sample ST/HNO$_3$ as determined from the XPS survey spectrum, includes the following elements: O, Al, Si and P. Al and O denote the formation of Al oxide, Si originates from the alloy and P originates from the chemical pre-treatment. After immersion in conversion bath ST650, the contents of Si, P and especially of Al, were reduced. In contrast, the content of Cr, Zr and F increased with a longer time immersion. Cr, Zr and F were not present on the pre-treated surface ST/HNO$_3$ and originate from conversion coating. With increasing conversion time, the Zr/Cr ratio increased from <1 to 9, the lowest being for 90 s. At the same time the ratio O/F decreased, indicating an increase content of F at longer conversion times. The Cr content on the surface after 11 and 18 min immersion was *cca.* 2.5 times smaller than the Zr content (Table 8). It is therefore reasonable to consider conversion coating ST650 to be a zirconate film that contains Cr [7,32], as will also be proven by high-resolution spectra (Figures 15 and 16).

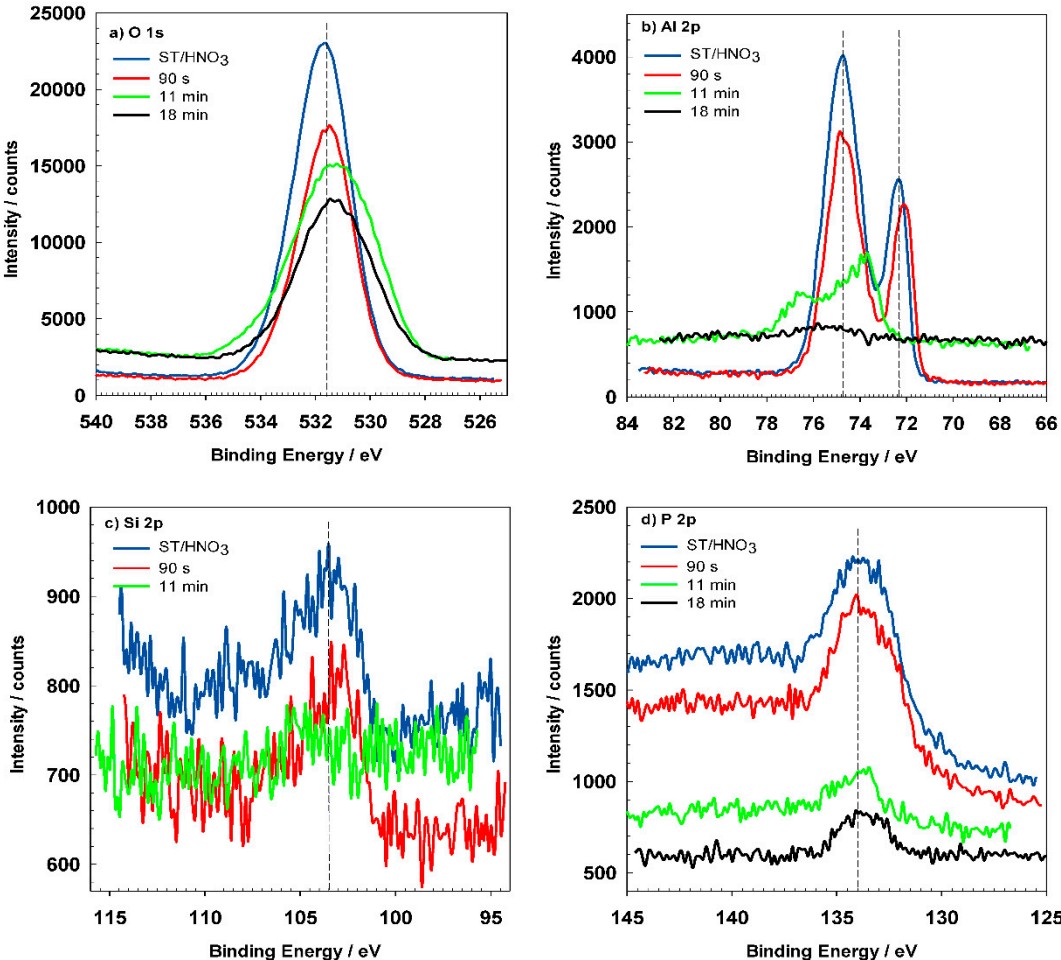

**Figure 15.** High-resolution XPS spectra of (**a**) O *1s*, (**b**) Al *2p*, (**c**) Si *2p* and (**d**) P *2p* recorded for AA3003 samples after chemical cleaning using ST/HNO$_3$ and then coated using 10% ST650 conversion coatings prepared at different conversion times: 90 s, 11 min and 18 min at room temperature. Spectra were not normalized on the intensity scale.

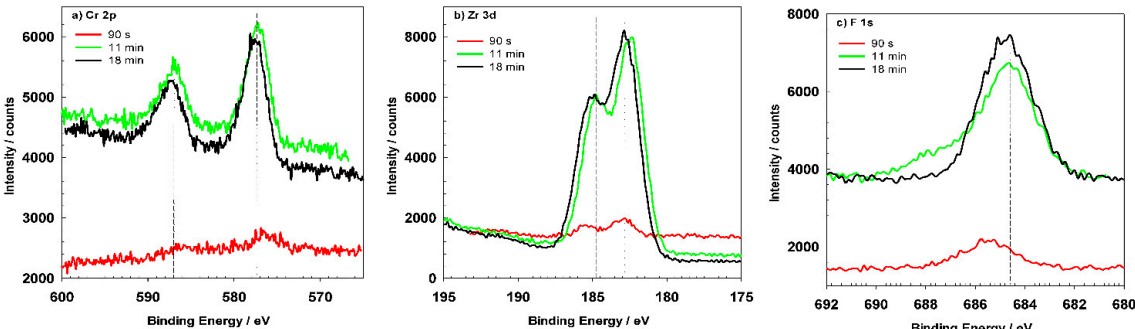

**Figure 16.** High-resolution XPS spectra of (**a**) Cr *2p*, (**b**) Zr *3d* and (**c**) F *1s* recorded for AA3003 samples after chemical treatment using ST/HNO₃ and then coated using 10% ST650 conversion coatings prepared at different conversion times: 90 s, 11 min and 18 min at room temperature. Spectra were not normalized on the intensity scale.

**Table 8.** The concentrations of elements obtained by XPS analysis of chemically cleaned AA3003 and after being coated with 10% ST650 conversion coatings prepared at different conversion times: 90 s, 11 min and 18 min.

| Sample | at.% O | at.% Al | at.% Si | at.% Cr | at.% Zr | at.% F | at.% P |
|---|---|---|---|---|---|---|---|
| ST/HNO₃ | 71.6 | 20.3 | 0.7 | - | - | - | 7.4 |
| 90 s ST650 | 64.8 | 19.1 | 1.4 | 1.2 | 0.7 | 2.1 | 10.7 |
| 11 min ST650 | 66.9 | 13.9 | 0.1 | 1.2 | 11.4 | 6.2 | 0.4 |
| 18 min ST650 | 66.4 | 5.6 | - | 1.8 | 16.3 | 8.4 | 1.5 |

Figure 15a depicts high-resolution peaks for O *1s* which is centered at 531.5 eV and corresponds to the hydrated Al oxide. Coated samples show similar position of the O *1s* peak but the shape of the peak became broader, indicating that was composed of several oxides, i.e., Al, Cr, and Zr. The Al *2p* spectrum (Figure 15b) for pre-treated AA3003 consist of two peaks at 72.3 eV and 74.8 eV characteristic for Al metal and oxidized Al [15]. After immersion in the conversion bath for 90 s, the Al *2p* peaks slightly shifted but the shape and Al content remained similar (Figure 15b). This indicates that the coating formation had just started and that it had not yet covered the Al surface (see Figure 16a,b). For the sample prepared for 11 min, Al the peaks were reduced, and diminished after 18 min (Table 8). Figure 15c represents the Si *2p* peak located at 103.4 eV, which was in accordance with the presence of SiO₂ [33]. It became negligible due to coverage of conversion coating. The intensity of P *2p* peak was also smaller for coated samples (Figure 15d). Its position at 133.8 eV can be related to phosphate present in the cleaning agent [33].

The most important elements related to conversion coatings are Cr and Zr (Figure 16). Cr, Zr and F could not be detected for coatings prepared at 90 s but only after longer conversion times. The shape of the spectra appears to be similar after 11 and 18 min, but the content of these elements increased for longer immersion times, consistent with the postulated mechanism of increased coating thickness with potential approaching the OCP vs. *t* plateau (Figure 5). Spectra for O *1s*, Cr *2p*, Zr *3d* and F *1s* recorded for coating prepared at 18 min conversion time were deconvoluted using selected component sub-peaks. Details on parameters used for deconvolution of experimental spectra are given in Appendix B (Table A3).

The Cr *2p* spectrum (Figure 17a) was deconvoluted using five component $2p_{3/2}$ and $2p_{1/2}$ doublets reflecting all compounds which may be formed in the conversion coating: Cr metal at 574.2 eV, $Cr_2O_3$ at 575.3 eV, $Cr(OH)_3$ at 577.4 eV, Cr(VI) species at 579.7 eV and $CrF_3$ at 580.0 eV (values of $E_b$ cited relate to $2p_{3/2}$ peak). TCP coatings were mainly composed of Cr(III) species, although the formation of Cr(VI) was also noted [6,24,29,34,35]. Since a strong fluorine peak indicated the formation of fluorides in addition to oxides (Table 8, Figure 15), the presence of $CrF_3$ should be taken into account as well. The interpretation of peaks is rather difficult because of the close binding energies of Cr(VI) (between 579.5 eV and 580.2 eV) and $CrF_3$ (between 579.3 eV and 580.8 eV) [33]. Therefore, in fluoride containing materials, such as ST650 conversion coatings, the distinction between Cr(VI) species and Cr(III) attributed to $CrF_3$ is difficult. The spectrum in Figure 17a deconvoluted using all five component peaks shows that the major Cr component of ST650 TCP coating was Cr(III) hydroxide, $Cr(OH)_3$, with some contribution of Cr(III) fluoride, $CrF_3$. The area fraction related to $Cr(OH)_3$ was about 82% and that of $CrF_3$ 18%. Component peaks related to Cr, $Cr_2O_3$ and Cr(VI) species dropped to zero during deconvolution. If the component sub-peak of $CrF_3$ was not taken into consideration in the deconvolution, then the peak related to Cr(VI) species was increased; the example is given in Appendix B (Figure A7). Therefore, according to the spectra presented, it can be stated that the coating comprised $Cr(OH)_3$ and $CrF_3$. The presence of a small amount or traces of Cr(VI) cannot be excluded but also cannot be confirmed with great accuracy.

The Zr $3d_{5/2}$ and $3d_{3/2}$ peaks were centered at 182.5 eV and 184.9 eV (Figure 17b). The experimental spectrum was deconvoluted using three-component $3d_{5/2}$ and $3d_{3/2}$ doublets reflecting all the compounds which may be formed in the conversion coating: Zr metal at 179.9 eV, $ZrO_2 \cdot 2H_2O$ at 182.5 eV and $ZrF_4$ at 183.5 eV (values of $E_b$ cited relate to $3d_{5/2}$ peak). The deconvoluted spectrum shows that the coating comprised mainly $ZrO_2 \cdot 2H_2O$, accounting for 66% of the peak area, and $ZrF_4$ with 34% of the peak area. It is noteworthy that fluorine can be present as fluoride, but also as oxyfluoride; however, due to the lack of appropriate reference compounds, we assumed the formation of fluoride.

Based on the results of the deconvolution of Cr *2p* and Zr *3d* spectra, the O *1s* spectrum was deconvoluted using three component peaks related to hydrated Cr(III) and Zr(IV) hydrated oxides, $Cr(OH)_3$ at 530.0 eV and $ZrO_2 \cdot 2H_2O$ at 531.5 eV, respectively (Figure 17c). The third component peak at 533.0 eV is related to bound water.

The F *1s* seemed to be comprised of two single peaks centered at 684.5 eV and 685.2 eV (Figure 17d). According to the XPS database [33], the $E_b$ values for $CrF_3$ were reported in the range 684.1–685.6 eV and that for $ZrF_4$ in the range 685.1–685.9 eV. These values are rather close but it seems that $CrF_3$ appeared at somewhat lower $E_b$. Therefore, we arbitrarily took component peaks for $CrF_3$ and $ZrF_4$ since, without appropriate standards, it is difficult to interpreted fluoride compounds (Appendix B, Table A3). According to the deconvolution of Cr *2p* and Zr *3d* spectra (Figure 17a,b), the contribution of $ZrF_4$ to the total peak was roughly double than that of $CrF_3$ (34% and 18%, respectively). Therefore, more fluorine was more bonded to Zr than to Cr (Figure 17 a,b). This was considered during deconvolution of F *1s* spectra by keeping the ratio of component peak at 1:2 (Figure 17d). The component peak of $CrF_3$ was centered at 683.9 eV and that of $ZrF_4$ at 685.1 eV.

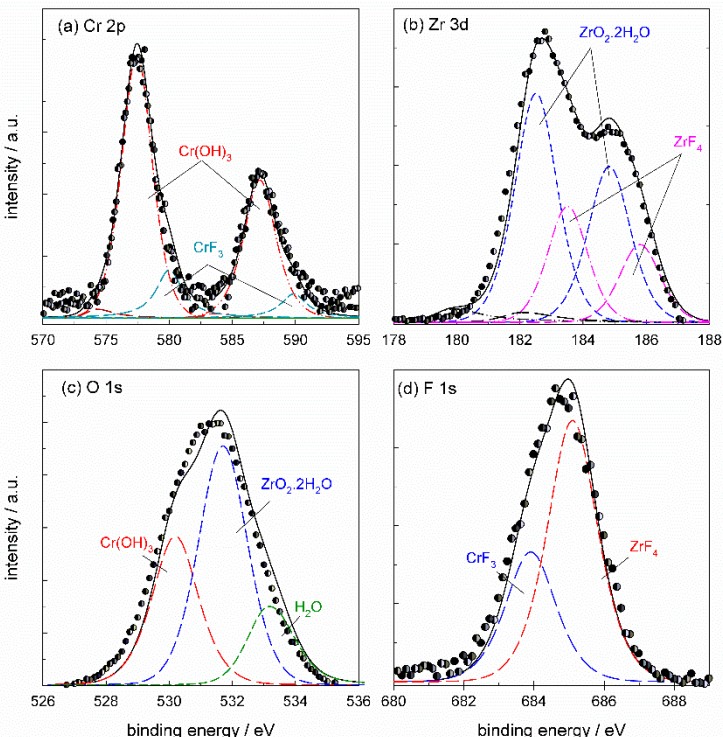

**Figure 17.** Deconvoluted high-resolution XPS spectra of (**a**) Cr *2p*, (**b**) Zr *3d*, (**c**) O *1s* and (**d**) F *1s* recorded for chemically pre-treated AA3003 sample coated using conversion coating ST650 prepared at 10% concentration and conversion time 18 min at room temperature. Experimental spectrum (symbols), fitted curve (solid line), component peaks (dashed lines).

ToF-SIMS depth profiles were recorded in order to reveal the in-depth composition and thickness of the conversion layer formed using 10 vol.% ST650 at 18-min conversion times (Figure 18).The most intense negative ion depth profiles were recorded: $ZrO_2^-$, $CrO_2^-$, $F_2^-$, $AlO^-$ and $Al_2^-$ and presented in the form of intensity vs. sputter time (Figure 18a) and normalized intensity vs. depth graphs (Figure 18b); please note that the sputtering rate was estimated as ~0.35 nm/s relative to a Ni/Cr standard. Three regions can be distinguished: conversion layer, interface layer and substrate. In the conversion layer, $ZrO_2^-$, $CrO_2^-$, $F_2^-$ and $AlO^-$ species prevailed, whilst in the interface layer, $AlO^-$, $Al_2^-$ and $F_2^-$ and species were dominant (Figure 18a,b). Taking into account the sputtering rate of 0.35 nm/s, the thickness of conversion layer was estimated to 40 nm, and that of the interface layer to 50 nm. These results are in line with data published for similar TCP coatings [2,7,34,36]. Looking at the depth profiles, especially with normalized intensities, it seems that the layer exhibited not only biphasic structure (conversion layer/interface layer) but even a tri-layer structure. Specifically, the outer part of the conversion coating seems to be enriched in Cr, and the inner part in Zr (Figure 18b). Similar results were noticed by Munson and Swain using AES depth profiles [36]. Such a structure is accompanied by a high fluorine content at the top and at the interface (Figure 18b).

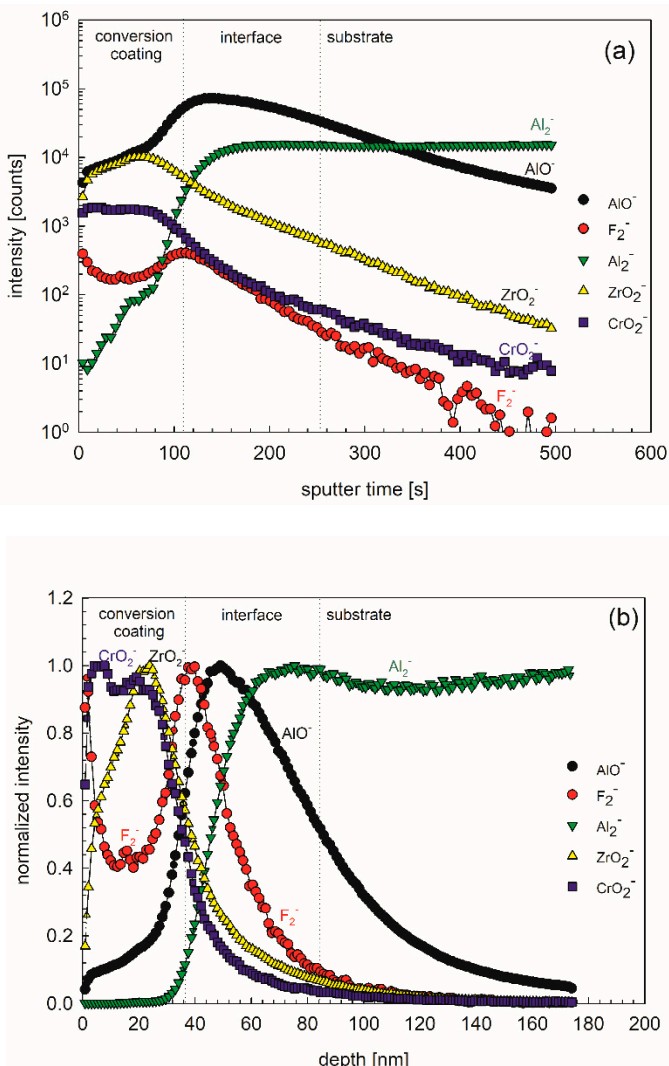

**Figure 18.** ToF-SIMS depth profiles (negative ions) recorded on the chemically pre-treated AA3003 sample coated using conversion coating ST650 prepared at 10% concentration and conversion time of 18 min at room temperature. Three regions are indicated, denoting conversion coating, interface layer and substrate. (**a**) presents intensity vs. sputter time and (**b**) normalized intensity vs. depth (sputter rate 0.5 nm/s relative to a Ni/Cr standard).

## 4. Conclusions

Commercial conversion coatings based on Zr and Cr (SurTec® 650) deposited on aluminum alloy 3003 were investigated by SEM/EDS and XPS analysis and electrochemical measurements in sodium chloride and simulated acid rain solutions. The following conclusions can be drawn:

1. Aluminum 3003 alloy has a heterogeneous structure with larger—a few micrometers sAl(Mn, Fe)—and smaller sub- and micrometer- sized Al(Mn, Fe)Si intermetallic particles. It exhibited a relatively good corrosion resistance in 3.5 wt.% NaCl and, especially, in simulated acid rain, where no localized breakdown was observed. With a longer immersion in these solutions (24 h), the corrosion resistance of AA3003 improved.

2. TPC SurTec® 650 coating containing hexafluoro zirconate and trivalent chromium was deposited at various concentrations and conversion times. The coating deposition followed the proposed mechanism of activation by dissolution of Al passive layer, pH-driven precipitation of Zr- and Cr-hydroxides and coating thickening.

3. At a constant conversion time of 18 min, increasing the concentration of the conversion SurTec 650® bath from 10 to 50 vol.% resulted in a change in coating morphology from uniform nodular (*cca*. 100 nm) to non-uniform and cracked. At the same time the Zr/Cr ratio decreased from 1.3–3.7 to 1.0–1.3 according to EDS and from 9.0 to 1.5 according to XPS. The discrepancy between the EDS and XPS results reflects the differences in analysis depth, which, for XPS, was related to the near-surface region. This may indicate a biphasic, or even tri-phasic coating structure, which was also noticed using ToF-SIMS depth profiles: the surface consists of a conversion coating rich in Zr and Cr, and an interface layer rich in $AlO^-$ and fluorine. The thickness of the conversion coating was estimated to 40 nm and that of the interface layer to 50 nm. The distribution of Zr and Cr within the conversion coating itself seems to change with depth: the outer part may be enriched in Cr, and the inner part in Zr.

4. At a constant concentration of the conversion SurTec® bath of 10 vol.%, increasing the conversion time from 90 s to 18 min resulted in an increase in the Zr/Cr ratio from 0.6 to 9.0 according to XPS. The coating prepared at 18 min conversion time contained mainly $ZrO_2 \cdot 2H_2O$ and some contribution of $Cr(OH)_3$. In addition to oxides, $ZrF_4$ and $CrF_3$ fluorides were formed. No Cr(VI) species could be detected by deconvolution.

5. Important results relate to the time dependence of the corrosion resistance of the coated AA3003 samples. None of the coated samples showed improvement of corrosion parameters after a short immersion time (15 min plus 1 h at OCP) regardless the electrolyte tested. However, after prolonged immersion (24 h plus 1 h at the OCP) significant improvement of corrosion performance occurred in both electrolytes—polarization curves for coated samples shifted in the positive direction and towards one order of magnitude smaller current densities.

6. The coating prepared from 10 vol.% SurTec® 650 at 18 min had a uniform nodular morphology without cracks. It exhibited the best corrosion performance in both sodium chloride and simulated acid rain, but only after prolonged immersion.

7. At prolonged immersion, the 10 vol.% coating acted as efficient anodic inhibitor due to the formation of a protective Zr/Cr hydroxide layer. Cathodic reaction was also diminished, indicating that oxygen reduction was also suppressed.

8. The reasons for the significant improvement of the corrosion behavior of both uncoated and coated samples will be investigated further in our next study. It seems that this behaviour was related to the changes within the coating as well as its interaction with the substrate.

9. It is noteworthy, however, that potentiodynamic measurements made after only 1 h immersion may not be relevant for judging the long-term behavior of the coated sample and may underestimate the coating performance, at least in the case of the AA3003 alloy.

**Author Contributions:** M.M.K.: preparation of the samples, electrochemical measurements and their analysis, figure preparation, writing—original draft preparation; B.K.: SEM/EDS analysis; U.T.: assistance in methodology and optimization of chemical pre-treatment; G.Š.: assistance in methodology of coating preparation; I.M.: conceptualization, supervision, deconvolution of XPS spectra, writing—review and editing.

**Funding:** This research was funded by Slovenian Research Agency, research core funding grant number P2-0393.

**Acknowledgments:** The authors thank Dolores Zimerl, for valuable technical assistance, and Maja Koblar, (Center for Electron Microscopy) for discussion regarding SEM/EDS analysis. The authors also acknowledge Janez Kovač and Tatjana Filipič, for XPS and ToF-SIMS measurements (Department of Surface Engineering and Optoelectronics).

**Conflicts of Interest:** The authors declare no conflict of interest.

## Appendix A

*Chemical Pre-Treatment*

To optimize the cleaning pre-treatment using commercial cleaners SurTec® 132 and SurTec® 089 and $HNO_3$ as a desmutting agent the following parameters were considered: (i) concentration of

SurTec® 132 and of SurTec® 089 to select the concentration within the mixture SurTec® 132/089, (ii) temperature and (iii) time immersion in SurTec® 132/089, and (iv) time immersion in HNO$_3$ solution (Figure A1). For the samples prepared under different conditions the potentiodynamic polarization curves in simulated acid rain were measured (Figures A2–A5, Table A1).

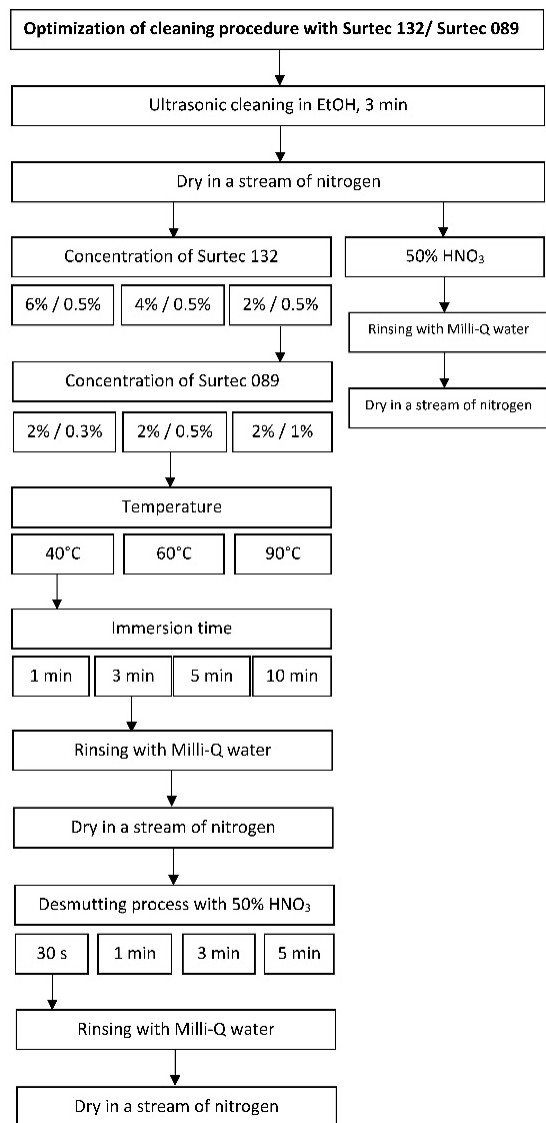

**Figure A1.** A schematic presentation of the selection of chemical pre-treatment of AA3003 substrate using commercial SurTec cleaners and HNO$_3$ as a desmutting agent.

The first step was to optimize the concentration of alkaline cleaners SurTec® 132 and SurTec® 089 at temperature of 40 °C and immersion time of 3 min (Figure A2, Table A1) giving the selected value 2% SurTec 132 and 0.5% SurTec® 089. For this concentration, the temperature of the mixture was tested at a constant immersion time of 3 min. The selected temperature of 40 °C exhibited the lowest $j_{corr}$ (Figure A3, Table A1). Next step was to select the immersion time in commercial SurTec cleaner (Figure A4, Table A1) and after that in 50% HNO$_3$ (Figure A5, Table A1) at constant concentration of cleaner and temperature.

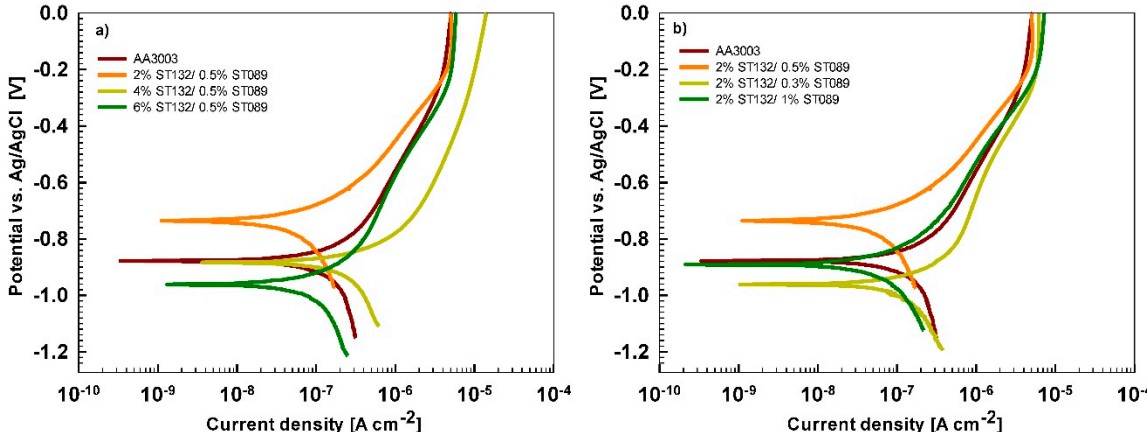

**Figure A2.** Potentiodynamic polarization curves recorded in simulated acid rain for AA3003 samples cleaned using different concentration of: (**a**) Surtec® 132; (**b**) Surtec® 089 at constant temperature 40 °C and time immersion 3 min.

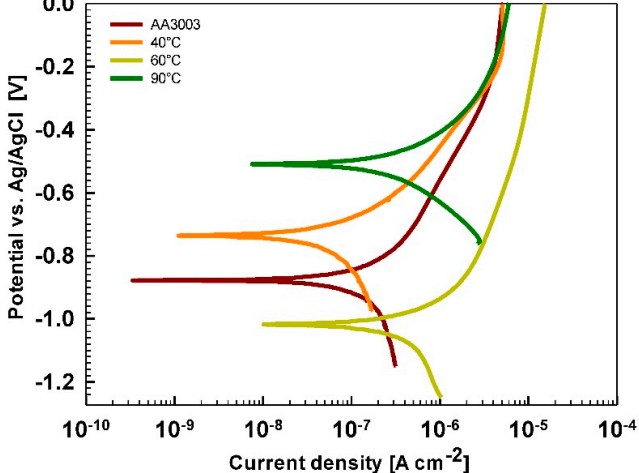

**Figure A3.** Potentiodynamic polarization curves recorded in simulated acid rain for AA3003 samples cleaned using 2% SurTec® 132 and 0.5% SurTec® 089 for 3 min at different temperatures.

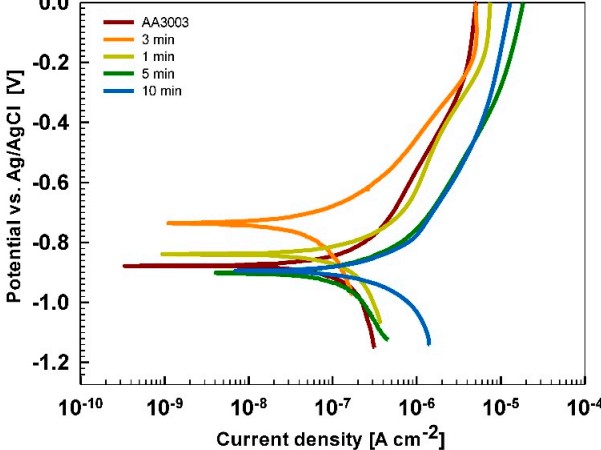

**Figure A4.** Potentiodynamic polarization curves recorded in simulated acid rain for AA3003 samples cleaned using 2% SurTec® 132 and 0.5% SurTec® 089 at 40 °C for different immersion times.

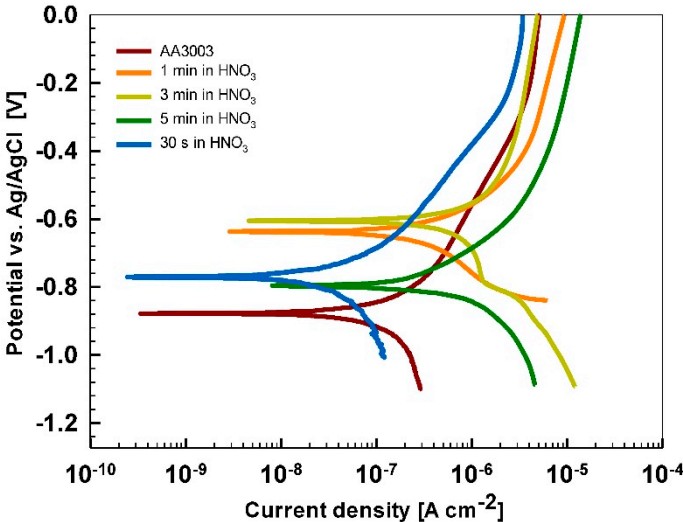

**Figure A5.** Potentiodynamic polarization curves recorded in simulated acid rain for AA3003 samples cleaned using 2% SurTec® 132 and 0.5% SurTec® 089 at 40 °C and immersion time of 3 min and desmutted in 50% HNO$_3$ for different desmutting times.

**Table A1.** Polarization resistance ($R_p$), corrosion current density ($j_{corr}$), and corrosion potential ($E_{corr}$) measured in simulated acid rain for as-received AA3003 and chemically pre-treated using ST/HNO$_3$.

| Parameter | Sample | $R_p$/kΩ cm$^2$ | $E_{corr}$/V | $j_{corr}$/µA cm$^{-2}$ |
|---|---|---|---|---|
| | AA3003 | 610 ± 131 | −0.88 ± 0.003 | 0.09 ± 0.02 |
| Concentration of ST132 | 2% ST132/ 0.5% ST089 | 854 ± 384 | −0.74 ± 0.001 | 0.06 ± 0.02 |
| | 4% ST132/ 0.5% ST089 | 106 ± 18 | −0.79 ± 0.13 | 0.30 ± 0.02 |
| | 6% ST132/ 0.5% ST089 | 575 ± 175 | −0.92 ± 0.05 | 0.08 ± 0.04 |
| Concentration of ST089 | 2% ST132/ 0.3% ST089 | 374 ± 16 | −0.78 ± 0.01 | 0.10 ± 0.01 |
| | 2% ST132/ 1% ST089 | 397 ± 226 | −0.93 ± 0.05 | 0.11 ± 0.04 |
| Temperature | 40 °C | 854 ± 383 | −0.74 ± 0.001 | 0.06 ± 0.02 |
| | 60 °C | 178 ± 131 | −0.86 ± 0.22 | 0.32 ± 0.03 |
| | 90 °C | 106 ± 18 | −0.47 ± 0.06 | 1.22 ± 0.04 |
| Immersion time | 3 min | 854 ± 383 | −0.74 ± 0.001 | 0.06 ± 0.02 |
| | 1 min | 215 ± 7 | −0.92 ± 0.11 | 0.18 ± 0.01 |
| | 5 min | 152 ± 50 | −0.96 ± 0.08 | 0.17 ± 0.04 |
| | 10 min | 216 ± 68 | −0.85 ± 0.06 | 0.35 ± 0.34 |
| Desmutting | 1 min in HNO$_3$ | 80 ± 50 | −0.62 ± 0.02 | 0.64 ± 0.47 |
| | 3 min in HNO$_3$ | 50 ± 1 | −0.60 ± 0.01 | 0.85 ± 0.02 |
| | 5 min in HNO$_3$ | 63 ± 8 | −0.71 ± 0.12 | 1.81 ± 0.07 |
| | 30 s in HNO$_3$ | 825 ± 229 | −0.76 ± 0.01 | 0.07 ± 0.02 |

Substrate pre-treatment beneficially influences the subsequent coating process as shown in Figure A6. Figure A6 shows the comparison between non-pre-treated and pre-treated AA3003 substrate coated with ST650 coating after 15 min and 24 h immersion in 3.5 wt.% NaCl. It is evident that better corrosion resistance was achieved on chemically pre-treated substrate.

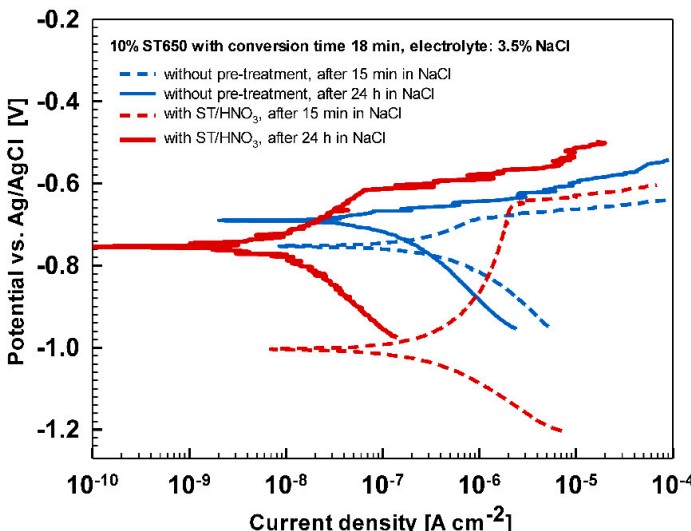

**Figure A6.** Potentiodynamic polarization curves recorded after 15 min and 24 h immersion in 3.5 wt.% NaCl for AA3003 samples coated with 10% ST650 coating prepared for 18 min at room temperature. Samples were prepared with and without chemical pre-treatment using ST/HNO$_3$. Electrochemical data deduced from the curves are presented in Table A2.

**Table A2.** Polarization resistance ($R_p$), corrosion potential ($E_{corr}$), corrosion current density ($j_{corr}$), breakdown potential ($E_{bd}$) and $\Delta E = |E_{bd} - E_{corr}|$ measured in 3.5 wt.% NaCl as a function of sample pre-treatment (Figure A6).

| Sample | Immersion Time | $R_p$/k$\Omega$ cm$^2$ | $E_{corr}$/V | $j_{corr}$/$\mu$A cm$^{-2}$ | $E_{bd}$/V | $\Delta E$/V |
|---|---|---|---|---|---|---|
| Without pre-treatment | 15 min | 70 ± 42 | −0.69 ± 0.01 | 0.19 ± 0.09 | - | - |
| With ST/HNO$_3$ | | 187 ± 96 | −0.95 ± 0.07 | 0.27 ± 0.09 | −0.65 ± 0.002 | 0.30 |
| Without pre-treatment | 24 h | 143 ± 80 | −0.688 ± 0.004 | 0.13 ± 0.04 | −0.69 ± 0.01 | 0.003 |
| With ST/HNO$_3$ | | 1927 ± 73 | −0.76 ± 0.01 | 0.01 ± 0.01 | −0.59 ± 0.04 | 0.17 |

## Appendix B

*Parameters Used for Deconvolution of XPS Spectra*

XPS spectra were background subtracted, using the non-linear, iterative Shirley method and processed by XFIT software developed by Wolff [37]. The latter is based on a least squares trial and error procedure with mixed Gaussian/Lorentzian peaks based on the method of Sherwood [38]. They were evaluated using standard peaks whose parameters, i.e., binding energy, E$_b$, width and Gaussian/Lorentzian ratio, were changed only minimally during the fitting procedure. The allowed variations of the binding energy and peak width are given in Table A3. Using this fitting procedure the height and area of contributing peaks can be determined, thus allowing the separation of measured spectra, Cr *2p*, Zr *3d*, O *1s* and F *1s*, to give the contributions of the different species.

The fitting procedure was conducted using reference component peaks determined experimentally. As reference standard, sputtered-cleaned metal samples and metal samples oxidized in oxygen atmosphere at elevated temperatures were used: Cr oxidized 1 h at 600 °C for Cr$_2$O$_3$ and CrO$_3$ standards [39], and Zr oxidized for 4 h at 450 °C for ZrO$_2$ standard [40]. The parameters of Cr(OH)$_3$ were obtained as the difference between the high-temperature standard peak of Cr$_2$O$_3$ and spectra recorded for the electrochemically oxidized sample [39]. During the fitting procedure the ratios between Cr *2p$_{3/2}$* and *2p$_{1/2}$* peaks and Zr *3d$_{5/2}$* and *3d$_{3/2}$* peaks were kept constant at 0.5 and 0.67, respectively. The parameters for peaks related to CrF$_3$ and ZrF$_4$ are taken in accordance with the XPS NIST database [33]. The ratio between the area of CrF$_3$ and ZrF$_4$ component peaks was kept 1:2, in

accordance with data given in Table 8. In Table A3 the range of parameters of binding energy, peak width and Lorentzian/Gaussian ratio used during deconvolution is presented.

**Table A3.** Range of parameter of binding energy, peak width ad Lorentzian/Gaussian ratio used for deconvolution of experimental spectra (Figure 17).

| Compound | Peak | Binding Energy Range/eV | Width Range/eV | Lorentzian/Gaussian Ratio |
|---|---|---|---|---|
| Cr metal | Cr $2p_{3/2}$ | 574.1–574.6 | 1.99–2.01 | 0.45–0.5 |
| $Cr_2O_3$ | Cr $2p_{3/2}$ | 575.3–575.4 | 2.24–2.26 | 0.9–1.0 |
| $Cr(OH)_3$ | Cr $2p_{3/2}$ | 577.0–577.1 | 2.57–2.59 | 0.9–1.0 |
| Cr(VI) | Cr $2p_{3/2}$ | 579.4–579.7 | 2.4–2.5 | 0.9–1.0 |
| $CrF_3$ | Cr $2p_{3/2}$ | 580.0–580.3 | 2.4–2.5 | 0.9–1.0 |
| Zr metal | Zr $3d_{5/2}$ | 179.7–179.9 | 1.44–1.46 | 0.4–0.6 |
| $ZrO_2{\cdot}2H_2O$ | Zr $3d_{5/2}$ | 182.3–183.7 | 1.55–1.65 | 0.4–0.6 |
| $ZrF_4$ | Zr $3d_{5/2}$ | 183.5–183.9 | 1.55–1.65 | 0.4–0.6 |
| $CrF_3$ | O $1s$ | 529.8–530.5 | 1.65–1.80 | 0.4–0.6 |
| $ZrF_4$ | O $1s$ | 531.0–532.0 | 1.65–1.90 | 0.4–0.6 |
| $H_2O$ | O $1s$ | 533.0–533.6 | 1.65–1.90 | 0.4–0.6 |
| $CrF_3$ | F $1s$ | 683.9–684.4 | 1.63–1.76 | 0.4–0.6 |
| $ZrF_4$ | F $1s$ | 685.0–685.4 | 1.63–1.76 | 0.4–0.6 |

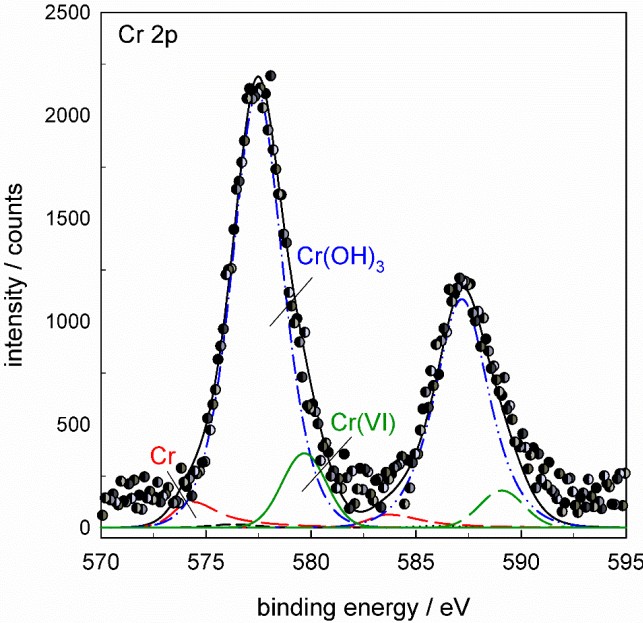

**Figure A7.** An example is given for the Cr *2p* spectrum deconvoluted using four component peaks: Cr, $Cr_2O_3$, $Cr(OH)_3$ and Cr(VI) (without $CrF_3$). Component peak related to Cr(VI) species dropped to zero when using component peak for $CrF_3$ (Figure 17a).

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
