# Peer review of "Protection of Aluminum Alloy 3003 in Sodium Chloride and Simulated Acid Rain Solutions by Commercial Conversion Coatings Containing Zr and Cr"

_coatings, doi:10.3390/coatings9090563_

Round 1
Reviewer 1 Report
Ref: coatings-568186 Title: Protection of aluminum alloy 3003 in sodium chloride and simulated acid rain solutions by commercial conversion coating containing Zr and Cr Journal: Coatings The paper entitled “Protection of aluminum alloy 3003 in sodium chloride and simulated acid rain solutions by commercial conversion coating containing Zr and Cr” reports some results in the field of metal protection. I hereby state my comments to the authors of the work mentioned above, hoping that they will be useful to enrich their work. I suggest it should be accepted after addressing the following questions. 1. The authors should provide the thickness of the conversion coating. 2. The authors should provide the compositions of the conversion coating by XRD.

Author Response
The authors should provide the thickness of the conversion coating.
ToF-SIMS depth profiles were made on sample which shows the best corrosion resistance (10 vol% ST650 and 18 min). The estimated thickness is 40 nm. New figure 18 is included in the manuscript and the description is added. In Experimental section details on ToF-SIMS analysis are added.
The authors should provide the compositions of the conversion coating by XRD.
In this study we used XPS and EDS to study the composition of the conversion layers. One could use also grazing incidence XRD to identity the composition of these thin layers. We will consider it for our future studies.
Reviewer 2 Report
The paper deals with the study of treated AA 3003 by a commercial product based on Zr and Cr. The main objective is to study the surface and investigate the alloys under chloride and simulated acid rain solutions. The authors compare and discuss differences in structural and morphological changes ongoing under different conditions. Comments:The paper is well written offering SEM, EDS, XPS and electrochemical study. However in my opinion some of the results are not contributive to the aim of the project. After alkaline and desmutting operation the samples were tested in chloride and simulated acid rain solutions. The study of the pre-treatment operations evaluated using electrochemical measurements does not contribute to fundamental comparison of the studied materials. It can be done by comparing AA 3003 and substrate covered with conversion coating that can show inhibiting action of the formed film. Although authors in text and in the conclusions mention the large penetration of conventional EDS measurements, it should be emphasized that the EDS results provide data but in the case of surface with intermetallic particles the results are serving as orientating. After forming a conversion layer the relevant methods used in the paper to characterize material is XPS method and electrochemical measurements. Concerning electrochemical measurements the fundamental output are Ecorr and ΔE. Presented standard deviations in the tables are doubtful. Why do the authors work with time of conversion layer formation before 18 min.? They conclude in the previous chapter that the conversion film is formed after 18 min. As well the results from electrochemical analysis confirm higher values of Ecorr in comparison to the others that can be attributed to the complete conversion layer formation. There are a few formal mistakes e.g.: zirconium hexafluoride does not exist; Table 2: Mn content in Location 3 is not correct. Rows 78-80: the sentence is confusing. After reading I advise major revision of this work with focus on the relevant results.
Author Response
English language and style are fine/minor spell check required
English language is corrected.
The paper deals with the study of treated AA 3003 by a commercial product based on Zr and Cr. The main objective is to study the surface and investigate the alloys under chloride and simulated acid rain solutions. The authors compare and discuss differences in structural and morphological changes ongoing under different conditions. Comments: The paper is well written offering SEM, EDS, XPS and electrochemical study. However in my opinion some of the results are not contributive to the aim of the project. After alkaline and desmutting operation the samples were tested in chloride and simulated acid rain solutions. The study of the pre-treatment operations evaluated using electrochemical measurements does not contribute to fundamental comparison of the studied materials. It can be done by comparing AA 3003 and substrate covered with conversion coating that can show inhibiting action of the formed film.
Reviewer is right, only the comparison of bare and covered substrate is relevant for the inhibiting action of the coating. The part devoted to alkaline and desmutting operation was optimized using electrochemical measurements in order to achieve optimal treatment. Substrate pre-treatment beneficially influences the subsequent coating process which is now shown in new Figure A6 added in Appendix. Figure A6 shows the comparison between non-pre-treated and pre-treated substrate coated with ST650 coating after 15 min and 24 h immersion in 3.5 wt.% NaCl. It is evident that better corrosion resistance was achieved on chemically pre-treated substrate. This is in the manuscript described in lines 590-595.
Although authors in text and in the conclusions mention the large penetration of conventional EDS measurements, it should be emphasized that the EDS results provide data but in the case of surface with intermetallic particles the results are serving as orientating. After forming a conversion layer the relevant methods used in the paper to characterize material is XPS method and electrochemical measurements.
Additional statement concerning the relevance of EDS analysis is now written (line 491).
Concerning electrochemical measurements the fundamental output are Ecorr and ΔE. Presented standard deviations in the tables are doubtful.
Standard deviation is now omitted for the values of DE as the latter were calculated from mean values of Ecorr and Ebd (Tables 3 and 6). Standard deviations given for vales of Ecorr are kept in the manuscript.
Why do the authors work with time of conversion layer formation before 18 min.? They conclude in the previous chapter that the conversion film is formed after 18 min. As well the results from electrochemical analysis confirm higher values of Ecorr in comparison to the others that can be attributed to the complete conversion layer formation.
Indeed, the shape of OCP vs. time curve during deposition of conversion coatings is well known from literature on other aluminium alloys. The establishment of plateau denotes the coverage of the surface by conversion layer. Since the literature data on AA3030 are scarce it was our opinion that we should check the formation as a function of conversion time as well. In my opinion it was worth it, especially for XPS analysis, since it showed the progressive formation of the coating.
There are a few formal mistakes e.g.: zirconium hexafluoride does not exist;
corrected to hexafluoro zirconic acid (line 47)
Table 2: Mn content in Location 3 is not correct.
corrected to 8.0 at%
Rows 78-80: the sentence is confusing.
The sentence is now written exactly as in the original reference by Guo and Frankel.
After reading I advise major revision of this work with focus on the relevant results.
The manuscript is corrected according to valuable comments given by the Reviewers.

Reviewer 3 Report
The paper possess high scientific and practical importance and can be published in present form with minor English stylistic corrections.
Author Response
English language and style are fine/minor spell check required
English language is corrected.
The paper possess high scientific and practical importance and can be published in present form with minor English stylistic corrections.
We thank Reviewer for this opinion.
Reviewer 4 Report
In this article, the authors studied the formation and optimization of a commercial conversion coating on AA3003. The coating primarily consists of Zr(IV) and Cr(III) oxides, which is a potential good replacement for conventional Cr(IV) conversion coating. The corrosion resistance of coated samples formed under various conditions was systematically investigated. The study is generally well designed and well presented. It is certainly an article worth considering for this journal and I recommend accepting it after the authors addressed the following minor comments:
Line 248-251: “the Ecorr was more negative and Ebd less positive resulting in an increase of delta E from 70 mV to 180 mV. The jcorr of the both samples were similar, but the passive range established for the chemically pre-treated sample increased compared to short immersion time (Figure 1a).”
The description does not match the data shown in Table 1.
Line 255: “Similar trend was observed”
In Fig 2b (24 h), the BT/HNO3 treated one showed a decrease of OCP, increased anodic current density, whist lower icorr, which are different to what are presented in Fig 2a (15 min), so they are not similar. Any further comments on the cause of these differences?
Line 268 “the micro-cracks formed at weak points”.
“Micro-cracks” is not a good term to use here because it indicates mechanical damage, which is not true here. “corrosion attack” is probably better. Also, the breakdown mechanism of Cl- anions on metal surface is still debatable. It may be more suitable to add some descriptions like “one possible mechanism may involve…”
Line 365: replace “more noble” with “more cathodic”. A simple example is that Mg has a very low Ecorr but it is not noble.
Fig 7b: any comment on why the 10% concentration caused anodic inhibition in simulated acid rain while it led to mixed inhibition in 3.5% NaCl?
Line 487-489: “Increased Al concentration observed at more concentrated coatings may be related to the underlying substrate which can be detected due to the presence of cracks within the coating, or Al may be partially incorporated in the coating”
This statement could be verified by looking at the oxidation state of Al in XPS spectra – if data is already available.
Fig 10-11: are the C.P.S. values off-set by a certain number?
Line 567: “results in faster depletion of the pit sites”
Not sure whether this is true – there’s no kinetics data regarding pitting corrosion shown here. Better to rephrase.
Line 626: Ration – typo
Fig 17: The fitting seems to make sense based on what could be present on the surface after the conversion process. My concern is that the peak area ratio does not seem to corelate well with the composition shown in Table 8. For example, the total concentration of Cr on the surface is 1.8. If 18% of Cr is CrF3, then the total concentration of CrF3 should be about 1.8 % *0.18 = 0.32 %. If we look at Fig 17d, the peak area for CrF3 and ZrF4 are similar, which means that about 50% of F should go into CrF3. In Table 8, the concentration of F is 8.4%. That means about 4% of F exists as CrF3 on the surface, which contradicts with the value 0.32% calculated with CrF3. It suggests that the F 1s peak should be re-fitted and the interpretation may need to be revised.
It is interesting that the conversion coating works well only after 24 hours of immersion instead of 15 min. It would be helpful if the authors could add some mechanistic descriptions regarding this phenomenon.
Author Response
English language and style are fine/minor spell check required
English language is corrected.
The paper possess high scientific and practical importance and can be published in present form with minor English stylistic corrections.
We thank Reviewer for this opinion.
Reviewer #4:
In this article, the authors studied the formation and optimization of a commercial conversion coating on AA3003. The coating primarily consists of Zr(IV) and Cr(III) oxides, which is a potential good replacement for conventional Cr(IV) conversion coating. The corrosion resistance of coated samples formed under various conditions was systematically investigated. The study is generally well designed and well presented. It is certainly an article worth considering for this journal and I recommend accepting it after the authors addressed the following minor comments:
Line 248-251: “the Ecorr was more negative and Ebd less positive resulting in an increase of delta E from 70 mV to 180 mV. The jcorr of the both samples were similar, but the passive range established for the chemically pre-treated sample increased compared to short immersion time (Figure 1a).” The description does not match the data shown in Table 1.
Indeed, the text was not completely precise and is now corrected.
Line 255: “Similar trend was observed”. In Fig 2b (24 h), the BT/HNO3 treated one showed a decrease of OCP, increased anodic current density, whist lower icorr, which are different to what are presented in Fig 2a (15 min), so they are not similar. Any further comments on the cause of these differences?
This part of the sentence (“similar trend was observed”) should have referred to Fig. 1. The text is now rephrased and it was pointed out that the scales of y-axes are different in Figures 2 and b. Actually, the anodic current density is smaller, as evident herein in the figure below, where curves recorded after 15 min and 24 h are given in the same graph.
In the manuscript, Figure 2 is now re-drawn to present y-axis on the same scale for (a) and (b): from -1.2 V to 0.6 V.
Line 268 “the micro-cracks formed at weak points”.“Micro-cracks” is not a good term to use here because it indicates mechanical damage, which is not true here. “corrosion attack” is probably better. Also, the breakdown mechanism of Cl- anions on metal surface is still debatable. It may be more suitable to add some descriptions like “one possible mechanism may involve…”
Reviewer is right. The text is corrected accordingly.
Line 365: replace “more noble” with “more cathodic”. A simple example is that Mg has a very low Ecorr but it is not noble.
corrected
Fig 7b: any comment on why the 10% concentration caused anodic inhibition in simulated acid rain while it led to mixed inhibition in 3.5% NaCl?
I believe that in both cases it causes mixed inhibition (please note that the scale in Figure 7b is expanded.
Line 487-489: “Increased Al concentration observed at more concentrated coatings may be related to the underlying substrate which can be detected due to the presence of cracks within the coating, or Al may be partially incorporated in the coating”. This statement could be verified by looking at the oxidation state of Al in XPS spectra – if data is already available.
When formed at 50 vol. % STA, the coating is nonhomogeneous (Fig. 9c) and the underlying substrate may be part of the analysed XPS spot (400 micrometres in diameter). The underlying substrate is oxidized due to pre-treatment so Al is present mainly in its oxidized state. Fitting of Al 2p spectra for 10% (left) and 50% (right) ST650-coated samples are given below. The component peak related to metal obtained by fitting is small and it is evident that in both cases Al(OH)3 component peak prevails. Due to low intensity of Al signal, the deconvolution is not very precise. In the manuscript, the following sentence is added: “Regardless the concentration of ST650, Al is present predominantly as Al hydroxide (results not shown).”
Fig 10-11: are the C.P.S. values off-set by a certain number?
As written in Experimental, all spectra were aligned by setting the C 1s peak to 248.8 eV.
Line 567: “results in faster depletion of the pit sites”. Not sure whether this is true – there’s no kinetics data regarding pitting corrosion shown here. Better to rephrase.
I agreed, rephrased.
Line 626: Ration – typo
corrected
Fig 17: The fitting seems to make sense based on what could be present on the surface after the conversion process. My concern is that the peak area ratio does not seem to corelate well with the composition shown in Table 8. For example, the total concentration of Cr on the surface is 1.8. If 18% of Cr is CrF3, then the total concentration of CrF3 should be about 1.8 % *0.18 = 0.32 %. If we look at Fig 17d, the peak area for CrF3 and ZrF4 are similar, which means that about 50% of F should go into CrF3. In Table 8, the concentration of F is 8.4%. That means about 4% of F exists as CrF3 on the surface, which contradicts with the value 0.32% calculated with CrF3. It suggests that the F 1s peak should be re-fitted and the interpretation may need to be revised.
The authors thank Reviewer for valuable comments. The fitting was corrected in such a way to keep the area ratio ZrF4/CrF3 component peaks at 2. As the Reviewer correctly suggested, according to deconvolution of Cr 2p and Zr 3d spectra and composition of the layer at% (Fig. 17 and Table 8), more fluorine is bonded to Zr than to Cr (ratio 2:1). The text in the manuscript is changed accordingly, Figure 17c is replaced and data in Table B1 corrected.
It is interesting that the conversion coating works well only after 24 hours of immersion instead of 15 min. It would be helpful if the authors could add some mechanistic descriptions regarding this phenomenon.
We find this behaviour very interesting and have studied it further using Zr-based only conversion coatings. This study will be submitted soon (G. Šekularac, I. Milošev, to be submitted) – reference 37 is added. As for now, we can state that this behaviour is related to the changes within the coating as well as its interaction with the substrate.
Reviewer 5 Report
The article entitled "Protection of aluminum alloy 3003 in sodium chloride and simulated acid rain solutions by commercial conversion coatings containing Zr and Cr" presents an extensive study on corrosion resistance on Al alloy. The text looks nice and I do not have any question about the quality of the manuscript.
However, there are some small faults, i.e. lines 261, 534 (Figure -> Figures), etc. Quality of some figures may be improved. The list of references looks poor, please add more if it is possible.
Author Response
The article entitled "Protection of aluminum alloy 3003 in sodium chloride and simulated acid rain solutions by commercial conversion coatings containing Zr and Cr" presents an extensive study on corrosion resistance on Al alloy. The text looks nice and I do not have any question about the quality of the manuscript.
However, there are some small faults, i.e. lines 261, 534 (Figure -> Figures), etc.
corrected
Quality of some figures may be improved.
It would be beneficial to suggest which figures. As suggested now I do not know which figure we are supposed to improve.
The list of references looks poor, please add more if it is possible.
Again, it would be beneficial to suggest which part of the presented introduction/results/discussion lacks the references. Additional reference is now included related to new ToF-SIMS data (ref. 36).
Round 2
Reviewer 2 Report
The paper can be accepted in the present form.